# Bacteria colonization in tumor microenvironment creates a favorable niche for immunogenic chemotherapy

See-Khai Lim[1], Wen-Ching Lin[1], Sin-Wei Huang[1], Yi-Chung Pan[1], Che-Wei Hu [ID][1], Chung-Yuan Mou[2], Che-Ming Jack Hu [ID][1][✉] & Kurt Yun Mou [ID][3]

## Abstract

**The tumor microenvironment (TME) presents differential selective pressure (DSP) that favors the growth of cancer cells, and monovalent therapy is often inadequate in reversing the cancer cell dominance in the TME. In this work, we introduce bacteria as a foreign species to the TME and explore combinatorial treatment strategies to alter DSP for tumor eradication. We show that cancer-selective chemotherapeutic agents and fasting can provide a strong selection pressure against tumor growth in the presence of bacteria. Moreover, we show that an immunogenic drug (oxaliplatin), but not a non-immunogenic one (5-FU), synergizes with the bacteria to activate both the innate and adaptive immunity in the TME, resulting in complete tumor remission and a sustained anti-tumor immunological memory in mice. The combination of oxaliplatin and bacteria greatly enhances the co-stimulatory and antigen-presenting molecules on antigen-presenting cells, which in turn bridge the cytotoxic T cells for cancer-cell killing. Our findings indicate that rational combination of bacterial therapy and immunogenic chemotherapy can promote anticancer immunity against the immunosuppressive TME.**

Keywords Bacteria Cancer Therapy; Immunotherapy; Differential Stress Resistance; Oxaliplatin; TME Remodeling
Subject Categories Cancer; Immunology; Microbiology, Virology & Host Pathogen Interaction

## Introduction

Solid tumors are a complex ecosystem composed of various cell types, including cancer cells, innate immune cells, adaptive immune cells, fibroblasts, or even bacteria (Chen and Song, 2022; Giraldo et al, 2019). In the tumor microenvironment (TME) cancer cells typically possess the highest fitness, which is acquired through their genome instability and highly proliferative nature through an microevolutionary process (Sanchez Alvarado, 2012; Vendramin et al, 2021). For example, it is known that cancer cells can suppress contact inhibition, elevate nutrient metabolism, and evade the surveillance of the adaptive immune system (Cornel et al, 2020; Gaikwad et al, 2022), thus leading to uncheck growth of tumor mass. Cancer cells can also manipulate other cells in the TME, such as educating innate immune cells or fibroblasts to become pro-tumor subtypes (e.g., M2 tumor-associated macrophages, myeloid-derived suppressor cells (MDSCs)) (Chen et al, 2019), which in turn contribute to cancer cell dominance in the TME.

Devising therapeutic strategies to disrupt the cancer dominance in the TME can draw inspirations from the competitive exclusion and niche displacement principles in ecological research, which shows that native species in an environment can be greatly threatened by an exotic species introduced from a completely different environment (Mooney and Cleland, 2001). The pioneering work of Dr. William Coley in the late 19th century has shown that a direct injection of live bacteria into sarcoma could lead to complete eradication of the tumor (Carlson et al, 2020), and modern studies revealed that several bacteria can specifically colonize tumors (Deb et al, 2022; Hosseini-Giv et al, 2021; Kim et al, 2023; Yamamoto et al, 2016) and reach a density as high as the cancer cells (~$10^8$ cells/gram tumor) (Weibel et al, 2008; Westphal et al, 2008). These works highlight the prospect of bacterial therapy toward displacing cancer cells as the dominant species in the TME. In addition, bacteria have been shown to activate the innate immune cells through their pathogen-associated molecular patterns (PAMPs), which can further modulate the evolutionary consequence of the TME. For example, lipopolysaccharide (LPS) from gram-negative bacteria can polarize macrophages toward the anti-tumor M1 subtype as well as prime antigen-presenting cells (APCs) for enhanced functionality of T-cell co-stimulation or cross presentation (Boutilier and Elsawa, 2021), thereby tipping the differential selective pressure against cancer cells. Despite the promise of bacterial therapy for retarding tumor growth, mouse experiments from our group and others showed that bacterial monotherapy rarely achieve complete tumor remission (Hu et al, 2022; Lehouritis et al, 2015; Murphy et al, 2017; Westphal et al, 2008). Combinatorial treatments that further modulate the

---

[1]Institute of Biomedical Sciences, Academia Sinica, Taipei 11529, Taiwan. [2]Department of Chemistry, National Taiwan University, Taipei 106319, Taiwan. [3]Deceased: Kurt Yun Mou passed away on August 28th, 2023. [✉]E-mail: chu@ibms.sinica.edu.tw

differential selective pressure (DSP) against cancer cells are thus desirable toward displacing the cancer cell dominance in the TME.

Growing advances of cancer biology and pharmaceutical development have led to evolving strategies to disrupt the equilibrium of the TME and exert increasing selective pressure against cancer cells. For instance, chemotherapeutics is typically designed to tackle specific molecular circuitry of cancer cells (Ramos et al, 2021; Zhou et al, 2020), and they can be rationally tailored to bestow survival advantages to competing species. Recent recognition of immunogenic chemotherapy showed that some cytotoxic agents, including doxorubicin, cyclophosphamide, and oxaliplatin, can enhance the population of tumor-infiltrating lymphocytes through the induction immunogenic cell death (ICD) (Obeid et al, 2007; Tesniere et al, 2010), enabling immune cell-mediated modulation of DSP in the TME. In addition, nutrient deprivation strategies such as fasting and intake of glucose-derivatives have been adopted to reduce tumor growth by tackling cancer cells' strong dependence on glucose metabolism (Naveed et al, 2014). These works highlight distinctive approaches to alter the DSP in the TME, presenting the possibility that rational integration of treatment strategies could heighten selective pressure against cancer cells.

In this work, we explore combinations of bacterial therapy with different treatment modalities toward establishing DSP that disfavors cancer cell survival. We show that several treatment strategies, including chemotherapy and fasting, can establish differential stress resistance (DSR) that bias the survival of bacteria over cancer cells in the TME. In particular, we observed that 5-fluorouracil (5-FU) and oxaliplatin, two common first-line drugs for colorectal cancer (CRC) patients, are selectively more toxic to cancer cells than to *E. coli*. In syngeneic mouse CRC models, the anti-tumor activity was only moderate when applying monotherapy of 5-FU or oxaliplatin alone, but dramatically enhanced when *E. coli* was co-administered. Of particular note, the immunogenic drug oxaliplatin in combination with *E. coli* completely eradicated all the tumors in the mouse model. Analyses of the tumor-infiltrating immune cells and the cultured immune cells revealed that the co-treatment of oxaliplatin and *E. coli* successfully stimulated both the innate and adaptive immune systems, and a favorable synergism was observed as the *E. coli* treatment reversed the immune-suppressive effect of oxaliplatin monotherapy.

# Results

## Anticancer differential selection pressure upon intratumoral bacterial therapy is enhanced by chemotherapy and fasting

We rationalize that upon bacteria colonization, a competitive relationship is established between the bacteria and the cancer cells in the TME. An effective therapy may be achieved by introducing an external selection pressure against the survival of the cancer cells but not the bacteria. 5-FU, a first-line treatment of CRC patients, is an uracil anti-metabolite, which inhibits DNA synthesis by blocking the thymidylate synthase (Longley et al, 2003). We assayed the cytotoxicity of 5-FU against the murine CRC cell line

MC38 and *E. coli* (DH5α). The $IC_{50}$ was 0.22 μM for MC38, which was >800-fold more toxic than that for *E. coli* ($IC_{50} = 339.1$ μM) (Fig. 1A,B). To assess the anti-tumor activity in vivo, the C57BL/6 mice were subcutaneously inoculated with the MC38 cells and randomly grouped for one of the four treatments: PBS, 5-FU, *E. coli*, and 5-FU + *E. coli*. All bacteria treatments were administered intratumorally unless otherwise specified. Compared to the PBS control, the monotherapy of 5-FU or *E. coli* moderately suppressed the tumor growth. (Fig. 1C). Intriguingly, the tumor progression was strongly inhibited by the co-treatment of 5-FU + *E. coli* (Fig. 1C,D). Similar trends were found in another CT26 CRC tumor model with BALB/c mice, which showed robust tumor suppression by the 5-FU + *E. coli* co-treatment (Appendix Fig. S1). Considering that short-term fasting after chemotherapy has been reported to selectively kill cancer cells due to the cells' higher nutrient demands and reduced adaptability to environmental cues (Naveed et al, 2014), we investigated nutrient deprivation as an alternative strategy for modulating DSR. With *E. coli* (DH5α) exhibiting greater resilience to nutrient deprivation compared to MC38 cells (Appendix Fig. S1C,D), we showed that in vivo nutrient deprivation through intermittent fasting modestly delayed tumor growth (Appendix Fig. S1E,F). However, as it is not feasible nor ethical to further restrict nutrient intake in the animal study for DSR modulation, we were prompted to examine other strategies for DSR enhancement.

## Combination of immunogenic chemotherapy and bacterial therapy eradicated established tumors while rescuing splenomegaly

With the aim of achieving complete tumor remission, we hypothesized that anticancer treatment can benefit from DSP modulation strategies that enhance anticancer immune cells in the TME, particularly adaptive immunity. Within the TME, tumor-infiltrating leukocytes tend to adopt a pro-tumoral phenotype. Although introducing *E. coli* into the TME can activate local innate immune responses, tumor-specific adaptive immunity is generally limited. Toward establishing DSR within the TME to target tumor cells, we explored an immunogenic drug with the ability to direct the host immune responses toward tumor cells. Examination of the TME of the 5-FU + *E. coli* group showed no enhancement of CD4[+] or CD8[+] tumor-infiltrating lymphocytes (TILs) (Appendix Fig. S3), indicating that the treatment combination was ineffective in elevating anticancer adaptive immunity (Vincent et al, 2010; Yamamura et al, 2015). We therefore identified oxaliplatin (OXA), another first-line chemotherapeutic agent for CRC treatment, to combine with bacterial therapy for two following reasons. First, unlike 5-FU, oxaliplatin is a potent immunogenic agent, which effectively induces ICD (Alcindor and Beauger, 2011; Tesniere et al, 2010). Combination of OXA and *E. coli* may thus elevate adaptive immune responses against tumor cells by presenting tumor-associated antigens to bacteria-activated immune cells. Second, it has been shown that the drug resistance of OXA was partially contributed by the M2 tumor-associated macrophages (TAMs) (Fu et al, 2019; Guo et al, 2017; Lan et al, 2021). We hypothesized that *E. coli* may strengthen the anticancer toxicity of OXA because LPS and other immunostimulants from *E. coli* are potent re-polarizers for the M2-to-M1 conversion of macrophages (Hu et al, 2022).

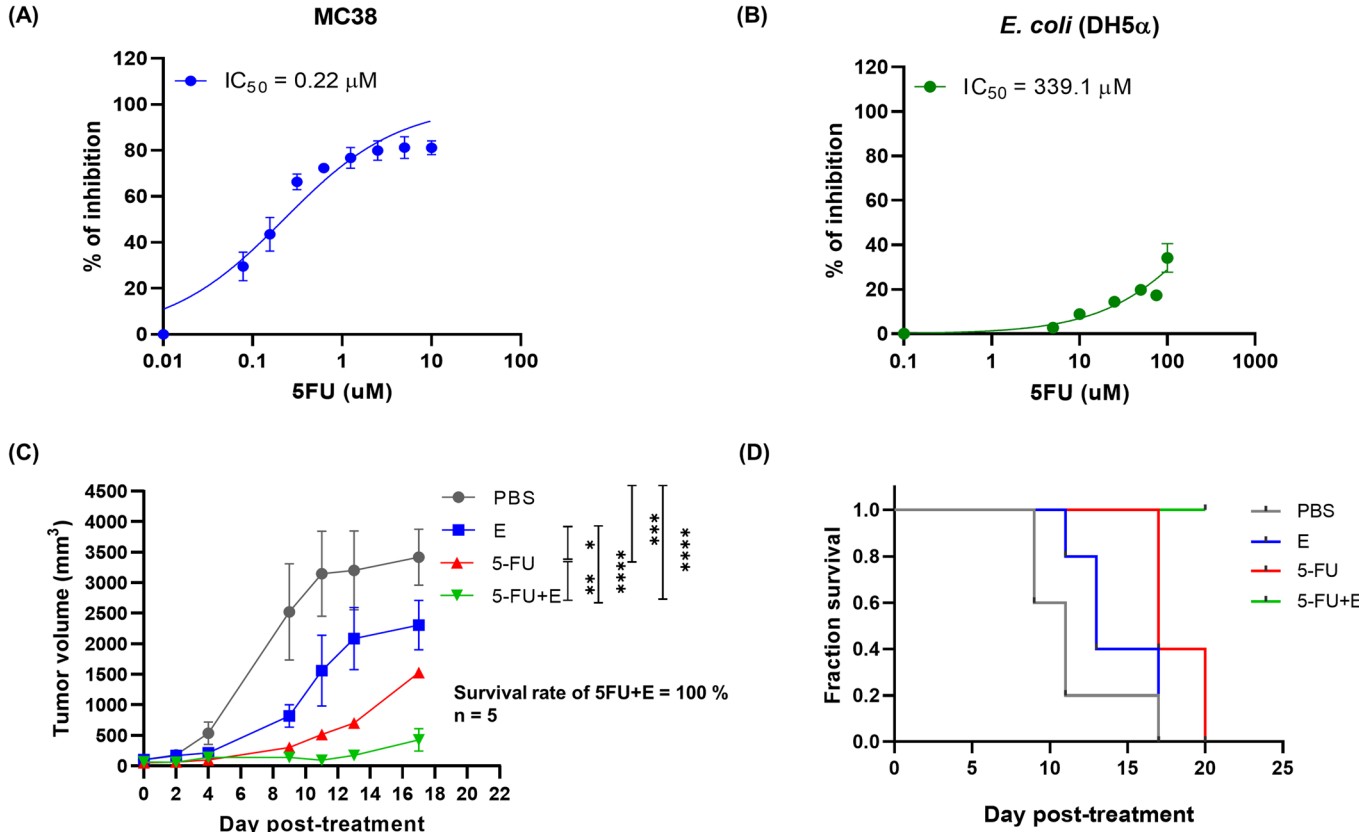

**Figure 1. 5-FU or starvation provided differential selection pressure against cancer cells over bacteria.**

(A) In vitro cytotoxicity of 5-FU against the CRC cancer cell line (MC38; $n = 3$). (B) In vitro cytotoxicity of 5-FU against the *E. coli* (DH5α; $n = 3$). (C) In vivo anti-tumor activity of PBS, 5-FU, *E. coli*, and 5-FU+*E. coli* in the MC38 syngeneic murine model, 5-FU and *E. coli* were delivered intraperitoneally and intratumorally, respectively ($n = 5$ mice for each treatment group). (D) Kaplan–Meier analysis of mouse survivals in (C). Data information: Mice were considered dead when the tumor volume exceeded 1500 mm³. All curves are presented as mean with SEM as error bars and all column plots are presented as individual value with SEM error bars. Statistical differences among tumor growth for more than 2 groups were computed using two-way ANOVA with Turkey's multiple comparison test. The differences were considered statistically significant when $p$-value <0.05 in all statistical analysis. Significance symbols are defined as *$p < 0.05$; **$p = 0.01$–0.05; ***$p = 0.0001$–0.001; and ****$p < 0.0001$. Source data are available online for this figure.

Similar to 5-FU, we found that OXA imposed a strong DSR against the cancer cells over *E. coli*, with an IC$_{50}$ of 2.2 μM for MC38 and IC$_{50}$ of 97.29 μM for *E. coli* (Fig. 2A,B). An additional assay comparing the number of tumor-harboring *E. coli* showed no significant differences between mice that received only *E. coli* or OXA + *E. coli*, indicating the in vivo dosage applied in our study did not affect the survival and colonizing ability of *E. coli* (Appendix Fig. S5C). Note that OXA is 10-fold less toxic than 5-FU (IC$_{50} = 0.22$ μM) against MC38, which was reflected in the mouse experiments as the OXA monotherapy (Fig. 2C) was less effective in suppressing tumor growth as compared to the 5-FU monotherapy (Fig. 1C). However, the co-treatment of OXA + *E. coli* greatly suppressed the tumor growth and eventually eradicated the tumors in all tested mice (CR = 5/5, Fig. 2C–E, and Appendix Fig. S4). We re-challenged the five cured mice with MC38 and observed complete rejection ($n = 2$) or significantly retarded tumor growth ($n = 3$), indicating establishment of anticancer immunological memory (Fig. 2F). To examine the bacterial therapy's clinical relevance for systemic cancer treatment and assess the *E coli*'s tumor-targeting ability, we treated the tumor-bearing mice through intravenous administration of *E. coli* and OXA. Similar to the i.t. treatment study, systemic

co-treatment of OXA + *E. coli* resulted in the most effective tumor suppression, followed by monotreatment of *E. coli*, OXA, and PBS (Fig. 2H, Appendix Fig. S5A). Of note, mice receiving either intratumoral or systemic *E. coli* exhibited no reduction in body weight (Appendix Figs. S2B and S5B). Peripheral blood analysis indicated a transient increase in neutrophils, which returned to normal by Day 7 post-treatment (Appendix Fig. S5E). The safety of the bacterial treatment was also confirmed by a hemolysis assay, which showed no hemolytic activity by the bacteria (Appendix Fig. S5D). Our data showed that the immunogenic chemotherapeutic drug OXA provided a better synergistic interaction with *E. coli* than the non-immunogenic 5-FU.

As splenomegaly, the enlargement of spleen caused by inflammation, is an adverse effect often associated with systemic bacterial therapy, we harvested the spleens and measured their weights on Day 15 after the initial treatments. As expected, the spleen size was enlarged in the mice treated with *E. coli* (Fig. 2G,I). Note that the *E. coli* treatment was delivered intratumorally, but the spleen was remotely affected. One mouse in the PBS group also experienced serious splenomegaly, which was presumably caused by the tumor burden (Mackay, 1965). Surprisingly, the OXA + *E.*

*coli* co-treatment rescued the spleen size to a normal range in all tested mice ($N = 5$), suggesting a reduction bacteria-associated systemic reactogenicity by the OXA combination.

## Co-treatment of OXA and *E. coli* overcomes OXA-mediated innate immune cell suppression in tumors

Upon analyzing the tumor-infiltrating immune cells in the four treatment groups, we observed that the viable tumor-infiltrating immune cells (CD45[+]) were highly suppressed by the OXA treatment, presumably due to the general cytotoxicity of chemotherapy (Figs. 3A and S8). Interestingly, the OXA + *E. coli* co-treatment completely reverted this negative effect, leading to a significant increase of viable immune cells as compared to the PBS control. The *E. coli* treatment also enhanced the viable immune cells to some extent in spite of the large individual variations. We further analyzed the activation state of these immune cells in tumors by intracellular staining of TNF-α and INF-γ. Remarkably, we found that only the co-treatment group, but not the *E. coli* or other groups, significantly activated the CD45[+] immune cells (Fig. 3B,C).

We next sought to focus on the tumor-infiltrating innate immune cells because PAMPs and DAMPs are potent immunostimulants to APCs. We first analyzed the tumor-associated neutrophils (TANs) since neutrophils are the first-line immune cells recruited to the infection site for the containment or clearance of bacteria. The monotherapy of either OXA or *E. coli* substantially elevated the TANs (Figs. 3D and S8), which typically possess a pro-tumor phenotype (Cupp et al, 2020; Ocana et al, 2017). Interestingly, the elevation of TANs was completely reversed in the OXA + *E. coli* group (Fig. 3D). We suspected that the suppression of TANs in the co-treatment group could be due to the elevated TNF-α, which is known to cause neutrophil death (Geering and Simon, 2011; Nandi and Behar, 2011; Salamone et al, 2001). Analogous to the M1/M2 classification in macrophages, neutrophils have also been categorized into the anti-tumor subtype N1 and the pro-tumor subtype N2 (Wang et al, 2018). Using TNF-α as the marker for differentiating N1/N2 subtypes, we found that most TANs in the PBS, OXA, and *E. coli* groups were N2-like (TNF-α⁻), but a significant number of N1-like TANs was induced in the OXA + *E. coli* group (Fig. 3E,F).

We next analyzed the TAMs, which are usually the most abundant immune cells in tumors. We found the population of TAMs was highest in the PBS group, followed by the *E. coli* group, the OXA + *E. coli* group, and the OXA group (Figs. 3G and S8). It is known that TAMs are often educated into the pro-tumor M2 subtype by the cues in the TME. Indeed, we found the TAMs in the PBS group were mostly M2-like (MHC-II⁻), and the OXA treatment did not alter this polarization (Fig. 3H). Encouragingly, despite a rather large variation among the individuals, the *E. coli* treatment significantly converted the TAMs into the M1-like subtype (MHC-II⁺). Moreover, the co-treatment of OXA + *E. coli* robustly re-polarized the majority of TAMs into the M1-like subtype (~90%). Strikingly, when analyzing the intracellular level of TNF-α and INF-γ, we found that only the TAMs in the co-treatment group were highly activated (Fig. 3I,J). It is worth noting that although the *E. coli* group showed elevated M1-like TAMs, these cells were either not activated or became dysfunctional as reflected by the lack of TNF-α and INF-γ upregulation.

## Adaptive immune cells were activated by the co-treatment of OXA and *E. coli*

Having shown the beneficial effects of the OXA + *E. coli* co-treatment on the innate immune cells, we next sought to investigate the treatment's influence on adaptive immunity. We analyzed CD4[+] and CD8[+] TILs and found only the OXA + *E. coli* group significantly increased the number of both T cell types (Fig. 4A,B, Appendix Figs. S8 and S11). This is in stark contrast to the 5-FU + *E. coli* treatment, where no enhancement of TILs was observed (Appendix Fig. S3). Similar to its negatory effect on macrophages, the OXA monotherapy severely depleted T cells in the TME (Fig. 4A,B). TNF-α and INF-γ staining revealed that only the OXA + *E. coli* co-treatment, but not the other three treatments, increased the number of functional CD8[+] TILs in the tumors (Fig. 4C,D). The elevated TILs in the OXA + *E. coli* group can be correlated with the tumor eradication and the establishment of the immunological memory induced by the combination treatment (Fig. 2F).

The overall compositions of tumor-infiltration immune cells are summarized in Fig. 4E. The TAMs were the most abundant immune cells in the PBS group, whereas the CD4+ and CD8+ TILs became highly dominant in the OXA + *E. coli* group (Fig. 4E). In terms of the TNF-α upregulation, all the immune cells investigated were stimulated by the OXA + *E. coli* co-treatment, but not the other three treatments (Figs. 3 and 4). It has been long reported that the neutrophil to lymphocyte ratio (NLR), usually calculated from peripheral blood samples but sometimes derived from infiltrating tissues such as tumors, is a useful indicator for cancer prognosis, a higher NLR implicates worse prognosis (Rao et al, 2012; Tokumaru et al, 2021). Figure 4F–H shows the NLRs of CD4[+], CD8[+], and the overall TILs for the different treatments. Both the monotreatment of OXA or *E. coli* showed very high NLRs because they either decreased the lymphocytes (OXA) or increased the neutrophils (*E. coli*). In stark contrast, the co-treatment of OXA + *E. coli* resulted in an extremely low NLR because it simultaneously decreased the neutrophils and increased the lymphocytes. Interestingly, the trend of NLRs (Oxaliplatin > *E. coli* > Oxaliplatin+*E. coli*) are directly correlated with the tumor progression (Oxaliplatin > *E. coli* > Oxaliplatin+*E. coli*) of the three groups.

## OXA and bacteria synergistically activate APCs and T cells in a cancer/immune cell co-culture system

Our in vivo studies demonstrated that OXA and *E. coli* synergistically modulate the immune milieu within the TME, shifting it from a pro-tumor state to an anti-tumoral state and thus exerting stronger selective pressure toward the tumor cells. To dissect the influence of OXA and *E. coli* on immune cells in contact with cancer cells under a controlled environment, we set up an in vitro assay where the murine splenocytes and the MC38 cancer cells were co-cultured and exposed to either individual treatments or a combination of OXA and *E. coli*. We first focused on the APCs, including B cells, dendritic cells (DCs), and macrophages, because they are directly responsive to DAMPs and PAMPs. We observed that all antigen-presenting cells (APCs), including B cells, are sensitive to treatment with either oxaliplatin (OXA) or *E. coli* (Fig. 5A). In both cases, APCs exhibited a consistent pattern: OXA

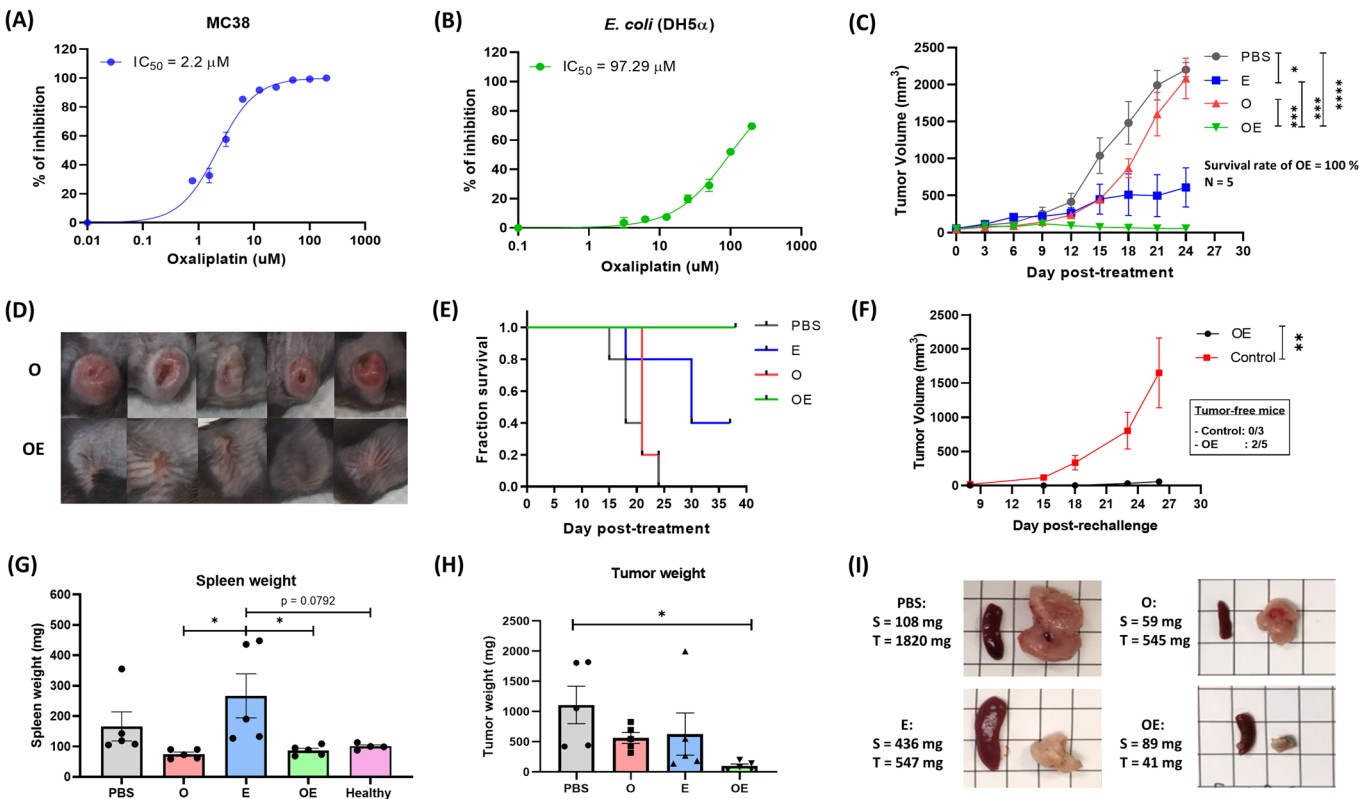

**Figure 2. The combination of OXA and *E. coli* eradicated the tumors without causing splenomegaly in mice.**

(A) In vitro cytotoxicity of oxaliplatin against the cancer cells (MC38; $n = 3$). (B) In vitro cytotoxicity of OXA against *E. coli* (DH5α; $n = 3$) (C) In vivo anti-tumor activity of PBS, OXA, *E. coli*, and OXA + *E. coli* in the MC38 syngeneic murine mode, OXA and *E. coli* were delivered intraperitoneally and intratumorally, respectively ($n = 5$ mice for each treatment group). (D) The appearances of tumors after 18 days of various treatments (complete figure are available in Appendix Fig. S4). (E) Kaplan–Meier analysis of mouse survivals in (C). (F) Tumor growth of the re-challenged mice versus the naïve mice ($n = 3$ and 5 for control and re-challenged mice, respectively; significance between two groups were computed using paired t-test). (G,H) spleen and tumor weight of each treatment group at Day 15 post-treatment ($n = 5$ for each treatment group and $n = 4$ for healthy control; significance among groups were computed using one-way ANOVA with Turkey's multiple comparison test). (I) Representative figure of spleens and tumors harvested from the mice in different treatment groups. Data information: Mice were considered dead when tumor volumes exceeded 1500 mm³. All curves are presented as mean with SEM as error bars and all column plots are presented as individual value with SEM error bars. Statistical differences among tumor growth for more than 2 groups were computed using two-way ANOVA with Turkey's multiple comparison test. The differences were considered statistically significant when p-value < 0.05 in all statistical analysis. Significance symbols are defined as *$p < 0.05$; **$p = 0.01$–$0.05$; ***$p = 0.0001$–$0.001$; and ****$p < 0.0001$. Source data are available online for this figure.

monotherapy (MO) led to a reduction in cell count across all APC types, while *E. coli* monotherapy (ME) increased the APCs compared to the MC38 control (M; Fig. 5A). Notably, among all APCs, B cells were the most sensitive to both treatments. To assess the functionality of APCs, we examined their ability to present exogenous antigens (MHC-II expression) and co-stimulate T cells (CD80+/CD86+ expression). As shown in Fig. 5B,C, co-incubation of splenocytes with MC38 slightly increased several activation markers, while the addition of *E. coli* as a treatment led to the most significant upregulation of both MHC-II and CD80/86 markers. In contrast, a slight decrease in the expression of these markers was observed in the oxaliplatin-treated group (MO) compared to the control group (M). This reduction can be attributed to the cytotoxicity of oxaliplatin toward immune cells, as evidenced by the lower cell counts shown in Fig. 5A.

In addition, the elevation in the co-stimulatory markers of APCs correlated with the activation of both CD4+ and CD8+ T cells, as demonstrated by the heightened expression of the CD25 surface

marker (Fig. 5D). Interestingly, the co-treatment of OXA + *E. coli* (MOE) effectively compensates for the cytotoxicity of oxaliplatin, as evidenced by the restoration of cell viability and the increased activation of APCs (MHC-II+ and CD80/86+) and T cells (CD25+) in comparison to the MC38 control (M). Moreover, to explore whether the immune cells were correlated with MC38-specific cytotoxicity, we conducted a parallel assay using the MC38-EGFP cell line (Fig. 5E). After a 24-h co-incubation period with each treatment, the viability of MC38-EGFP cells was assessed using a fixable viability dye Efluor780 (Thermo Fisher Scientific), which specifically stains dead cells. Intriguingly, despite the higher number of activated immune cells observed in *E. coli*-treated splenocytes, the combined treatment group (MOE) exhibited the most robust cytotoxicity against MC38-EGFP cells (15.10% and 32.27%, respectively; Fig. 5E). This finding suggests a potential role of oxaliplatin in inducing cancer-specific immunity within our experimental context (Appendix Fig. S15). The consistency between our in vitro and in vivo data highlights the potential of the

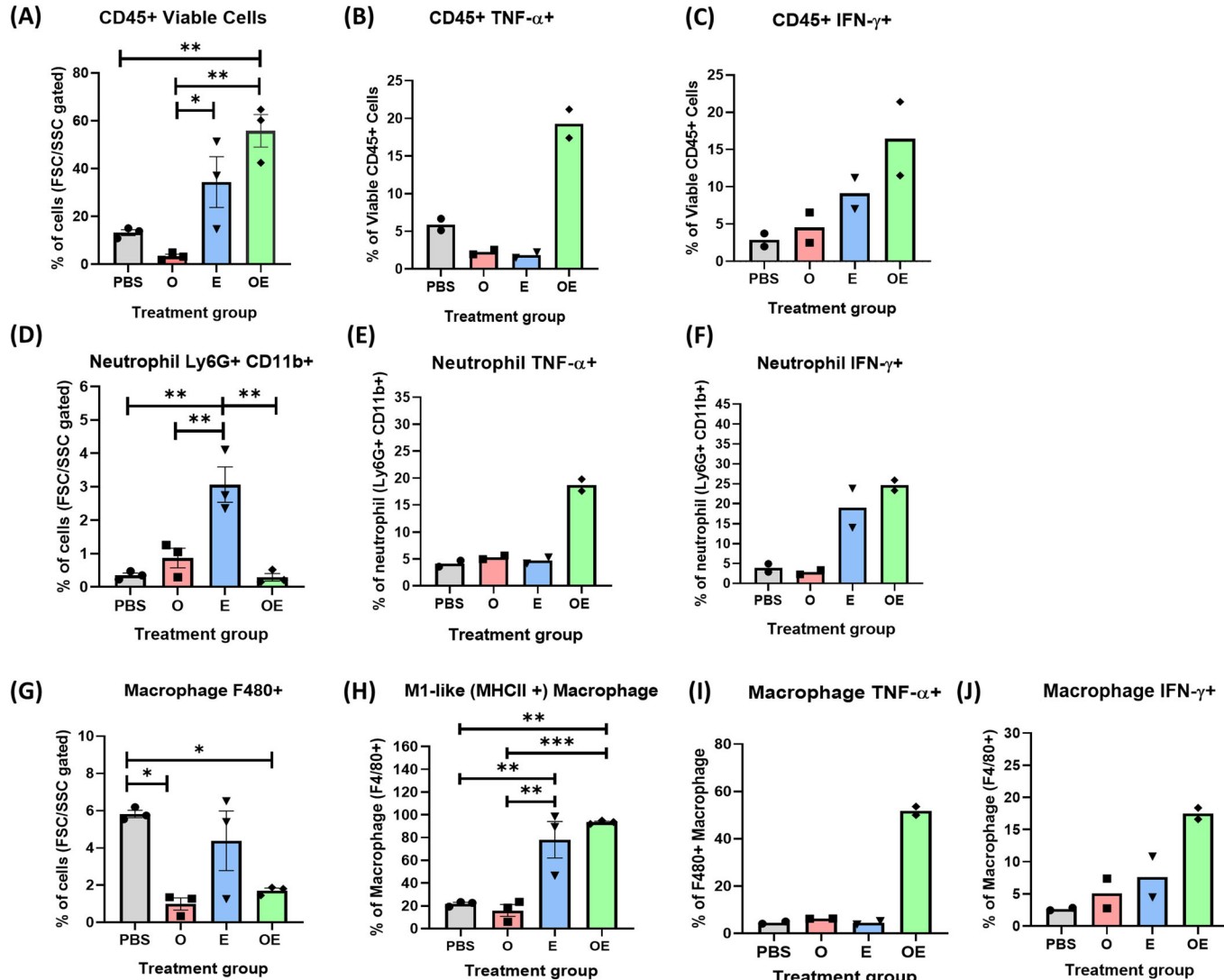

**Figure 3. Co-treatment of OXA + *E. coli* activated the tumor-associated macrophages and tumor-associated neutrophils.**

(A) Percentages of total viable tumor-infiltrating CD45+ immune cells. (B) Percentages of TNF-α+ cells in the viable tumor-infiltrating CD45+ immune cells. (C) Percentages of IFN-γ+ cells in the viable tumor-infiltrating CD45+ immune cells. (D) Percentages of tumor-associated neutrophils (Ly6G+CD11B+). (E) Percentages of TNF-α+ cells in the TANs. (F) Percentages of IFN-γ+ cells in the TANs. (G) Percentages of tumor-associated macrophages (F4/80+). (H) Percentages of M1-like TAMs (MHC-II+). (I) Percentages of TNF-α+ cells in the TAMs. (J) Percentages of IFN-γ+ cells in the TAMs. Data information: The sample size (*n*) is 3 for each group in surface staining and 2 for intracellular staining. All data were presented as individual data plots and error bars were presented as SEM of the data set if applied. Statistical differences among groups were computed (when *n* = 3) using one-way ANOVA with Turkey's multiple comparison test, the differences were considered statistically significance when *p*-value is <0.05. Gating strategies are available in Appendix Fig. S6 and representative flow cytometry figures are available in Appendix (Appendix Figs. S8, S9 and S10 for surface staining, TNF-α and IFN-γ intracellular staining, respectively). Significance symbols are defined as *$p < 0.05$; **$p = 0.01$–$0.05$; ***$p = 0.0001$–$0.001$; and ****$p < 0.0001$. Source data are available online for this figure.

OXA + *E. coli* co-treatment to activate immune cells, including APCs and T cells, thereby instigating tumor-specific immunity and tumor rejection. On a related note, the synergistic effect between oxaliplatin and *E. coli* was only observed when whole bacteria were administered. Co-incubation of OXA with LPS (100 and 1000 ng/mL) did not result in any alteration in the IC$_{50}$ profile of OXA (Appendix Fig. S13). To further verify the involvement of immune cells in cancer cell killing under the co-treatment condition, we labeled splenocytes with a fluorescent dye and visualized the co-culture by fluorescence microscopy. For the experiments without treatment or with either

OXA or *E. coli* as monotreatment, we found adherence of individual splenocytes to cancer cells (Appendix Fig. S14A). In stark contrast, the co-treatment induced large immune cell clumps aggregating on cancer cells or on killed cellular debris (Appendix Fig. S14A,B), indicating active engagement of immune cells with cancer cells. These immune clusters were comprised of both APCs and T cells (Appendix Fig. S16), which may collectively contribute to cancer killing. Altogether, the results highlight that OXA + *E. coli* treatment effectively mounts immune cell-mediated DSR for tumor inhibition.

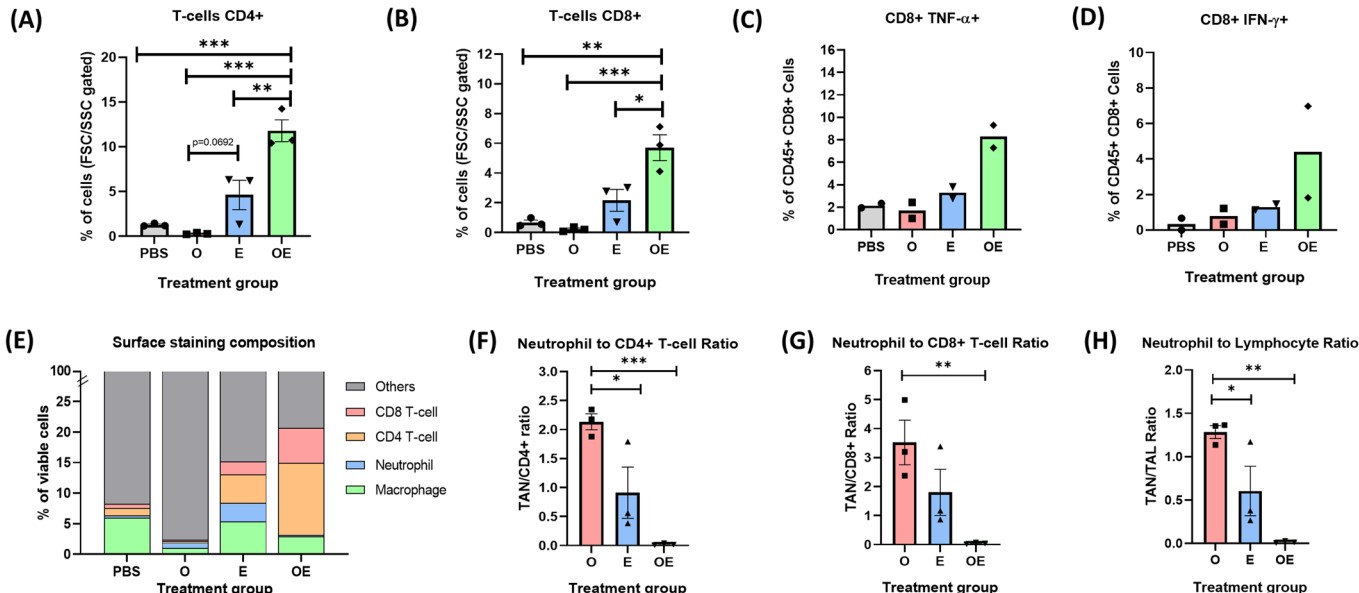

**Figure 4. Co-treatment of OXA + E. coli increased CD4⁺ and CD8⁺ tumor-infiltrating lymphocytes and lowered the neutrophil to lymphocyte ratio in tumors.**

(A) Percentages of CD4⁺ TILs. (B) Percentages of CD8⁺ TILs. (C) Percentages of TNF-α⁺ cells in the CD8⁺ TILs. (D) Percentages of IFN-γ⁺ cells in the CD8⁺ TILs. (E) Comparison of tumor-infiltrating immune cells. (F) Intratumoral NLR of CD4⁺ T cells. (G) Intratumoral NLR of CD8⁺ T cells. (H) Intratumoral NLR of total T cells. Data information: The sample size (n) is 3 for each group in surface staining and 2 for intracellular staining. All data were presented as individual data plots and error bars were presented as SEM of the data set if applied. Statistical differences among groups were computed (when n = 3) using one-way ANOVA with Turkey's multiple comparison test, the differences were considered statistically significance when p-value is <0.05. Gating strategies are available in Appendix Fig. S6 and representative flow cytometry figures are available in supplementary data (Appendix Figs. S8, S9 and S10 for surface staining, TNF-α and IFN-γ intracellular staining, respectively). Significance symbols are defined as *p < 0.05; **p = 0.01–0.05; ***p = 0.0001–0.001; and ****p < 0.0001. Source data are available online for this figure.

## Discussion

Solid tumors can be seen as ecosystems where the microenvironment is optimized for tumor cell growth. In ecological terms, tumor cells are the dominant species with limited competition from other members within the TME. In attempting to break this dominance, we introduced an invasive species (E. coli) into the TME to compete for survival resources. Our study shows that bacteria therapy can benefit from treatment combinations that differentially favor bacteria survival over tumors, such as cancer-selective chemotherapeutics and fasting. In particular, we observed profound synergism between bacterial therapy and an immunogenic drug, oxaliplatin. OXA combination with bacterial therapy converts the immunosuppressive TME to an immunoreactive one (Tie et al, 2022; Tormoen et al, 2018). Figure 6 illustrates the stepwise conversion of TME's selective factors from pro-tumor to those detrimental to the tumor cells. In the absence of external intervention, cancer cells dominate the TME, continuously suppressing immune cell activity and creating a highly immunosuppressive environment. Introduction of bacteria as an exogenous species induces a competitive environment that suppresses tumor growth, and chemotherapeutic agents as well as nutrient deprivation adds to the DSP against cancer. Combination of bacterial therapy with an ICD inducer, effectively releases tumor antigens and DAMPs, connecting bacteria-primed immune cells to the initiation of cancer-specific immunity. This synergy relies on E. coli's ability to activate tumor-infiltrating immune cells, particularly APCs, in conjunction with

OXA's induction of tumor-related DAMPs. These DAMPs can be taken up by the activated APCs and subsequently presented to the host adaptive immune system, leading to a more efficient generation of tumor-specific immune responses that eliminate tumor cells (Groza et al, 2018).

In addition to the immunostimulatory benefits of the combination therapy mentioned above, the chemotherapy and the bacterial therapy may also mutually enhance each other's effectiveness in several ways (Iida et al, 2013). Previous studies have suggested that the limited efficacy of bacterial therapy was partially attributed to the anti-pathogen defense from the innate immunity. For example, a neutrophil barrier was observed in the bacteria-treated tumors in mice, restricting the bacteria distribution to the central necrotic region (Westphal et al, 2008). Similarly, it was reported that the inoculation of E. coli in mouse tumors resulted in the redistribution of TAMs that phagocytosed E. coli and separated them from the proliferative rims of the tumors (Weibel et al, 2008). In our combination therapy, we observed that OXA effectively reduced the populations of TANs and TAMs, thus potentially favoring the bacterial growth within tumors (an example where chemotherapy helps the bacterial therapy). On the other hand, several studies have reported that the M2-polarized TAMs contribute to the OXA resistance by suppressing the cancer cell apoptosis (Fu et al, 2019; Guo et al, 2017; Lan et al, 2021). As expected, OXA has demonstrated synergy with R848, a small molecule that converts macrophages from M2 to M1, in cancer treatments (Li et al, 2021; Liu et al, 2020). Our results showed that the Oxa+E. coli

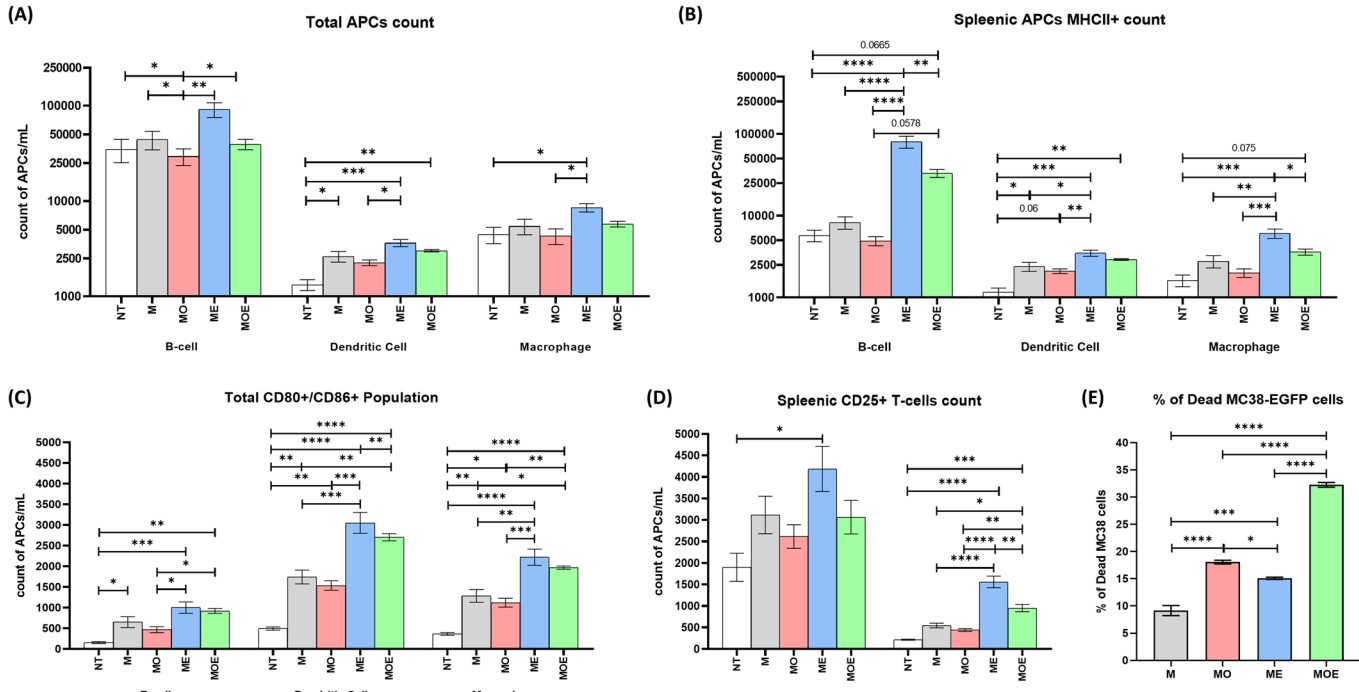

**Figure 5. The cancer/immune/bacteria co-culture system reveals the synergistic interaction of OXA and *E. coli* for immune cell activation.**

(A–E) The immune cells (murine splenocytes) and the cancer cells (MC38) were co-cultured and subject to various treatments (NT: splenocytes alone. M: splenocytes + MC38. ME: splenocytes + MC38 + *E. coli*. MO: splenocytes + MC38 + oxaliplatin. MOE: splenocytes + MC38 + oxaliplatin + *E. coli*). The medium contained penicillin-streptomycin to prevent the overgrowth of *E. coli* that led to medium acidification and cell death (Appendix Fig. S12), and the OXA concentration was kept at the $IC_{50}$ against MC38 (2 μM). The immune cells were analyzed by flow cytometry after a 24-h incubation period. (A) The populations of the three APCs (B cells, DCs, and macrophages) in the CD45+ cells. (B) The MHC-II$^+$ populations of the three APCs in the CD45+ cells. (C) The CD80$^+$/CD86$^+$ double positive populations of the three APCs in the CD45+ cells. (D) The CD25$^+$ population in CD8$^+$ T cells. (E) Percentage of the dead MC38-EGFP cells after receiving each treatment in a splenocyte co-culture system. Data information: All data were presented as individual data plots with SEM error bars. Statistical differences between among groups were computed using one-way ANOVA with Turkey's multiple comparison test and differences were considered statistically significance when *p*-value is <0.05 (*n* = 3 for each treatment group). Significance symbols are defined as \*$p < 0.05$; \*\*$p = 0.01$–0.05; \*\*\*$p = 0.0001$–0.001; and \*\*\*\*$p < 0.0001$. Source data are available online for this figure.

co-treatment robustly re-polarized the TAMs into the M1 subtype. Comparing the OXA monotherapy and the Oxa+*E. coli* combination therapy (Fig. 2C–E), it was clear that the drug resistance was developed after Day 12 in the OXA monotherapy, but the anticancer effect of OXA + *E. coli* persisted to the end of treatment (Day 24). This exemplifies a scenario where bacterial therapy aids the chemotherapy therapy in circumventing the drug resistance issue.

In summary, our study provides experimental evidence that bacterial therapy, an emerging novel modality in clinical trials, greatly synergizes with the well-established chemotherapy for cancer treatments. Our findings demonstrate that the co-administration of OXA and *E. coli* effectively eradicates the tumors in the syngeneic mouse models through several mechanisms, including the DSP that disfavors the cancer growth, the activation of both the innate and adaptive immune systems, and the alleviation of the drug resistance. Comparing the therapeutic efficacy and the immune profiles of tumors using 5-FU or OXA co-treated with *E. coli*, our results highlight the importance of a judicious choice of the combination. A vast combinatorial space with various chemotherapeutic agents and bacteria in different cancer settings is therefore ready for exploration.

# Methods

## Chemicals and reagents

The chemicals and reagents used in this research project is commercially available as follows: Oxaliplatin (Cayman Chemical; Cat# 13106), CFSE Fluorogenic esterase substrate (Abcam; ab145291), RBC lysis buffer (Biolegend; Cat# 420301), human IL-2 (PeproTech; Cat# 200-02), mouse IL-7 (SinoBiological; Cat# 50217-MNAE). For flow cytometry (fluorophore-conjugated anti-mouse antibodies): eFluor™ 780 eBioscience™ Fixable Viability Dye (Thermofisher Scientific; Cat# 65-0865; 1:2000), Pacific Blue™ CD45 antibody (Biolegend; Cat# 103125; 1:200), PerCP/Cyanine5.5 CD45 antibody (Biolegend; Cat# 103132; 1:200), APC CD4 antibody (Biolegend; Cat# 100516; 1:100), AF488 CD8a antibody (Biolegend; Cat# 100723; 1:200), Alexa Fluor® 647 Ly6G antibody (Biolegend; Cat# 127610; 1:200), FITC CD11B antibody (Biolegend; Cat# 101206 1:200), Pacific Blue CD11B antibody (Biolegend; Cat# 101224; 1:200), Brilliant Violet 421 F4/80 antibody (Biolegend; Cat# 123131; 1:50), AF700 F4/80 antibody (Biolegend; Cat# 123130; 1:200), AF700 CD19 antibody (Biolegend; Cat# 115528; 1:200), AF594 CD11C antibody (Biolegend; Cat# 117346; 1:100), FITC MHCII antibody (Biolegend; Cat# 107606; 1:100), BV421

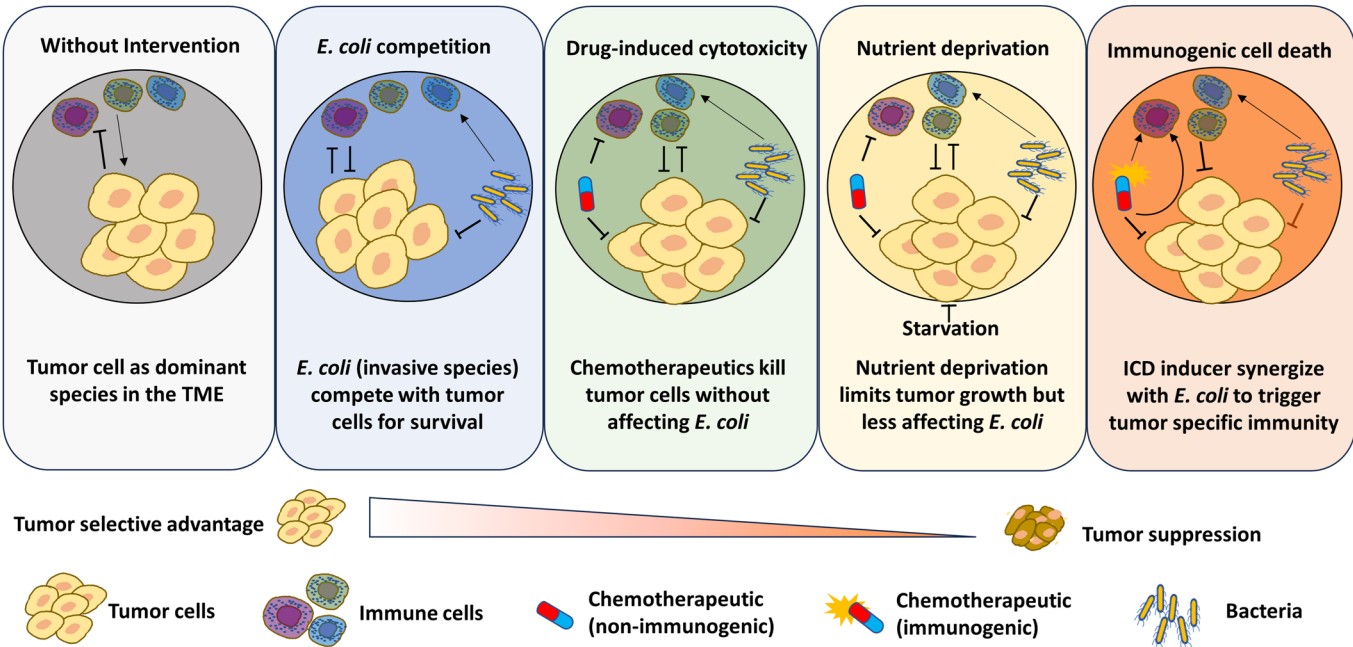

## Differential Selective Pressure

**Without Intervention**

Tumor cell as dominant species in the TME

***E. coli* competition**

*E. coli* (invasive species) compete with tumor cells for survival

**Drug-induced cytotoxicity**

Chemotherapeutics kill tumor cells without affecting *E. coli*

**Nutrient deprivation**

Starvation

Nutrient deprivation limits tumor growth but less affecting *E. coli*

**Immunogenic cell death**

ICD inducer synergize with *E. coli* to trigger tumor specific immunity

Tumor selective advantage — Tumor suppression

Tumor cells Immune cells Chemotherapeutic (non-immunogenic) Chemotherapeutic (immunogenic) Bacteria

**Figure 6. Effects of bacteria and chemotherapeutics toward cancer cells and how they shaped the tumor microenvironment.**

(i) Tumor without medical intervention. The cancer cells suppress the immune cells or even educate the immune cells to assist the tumor progression. (ii) Tumor with chemotherapy. The chemotherapeutic agent non-selectively killed cancer cells and immune cells. (iii) Tumor with bacterial therapy. The innate immune cells are activated by PAMPs. The cancer growth is also impeded due to the competition from bacteria. (iv) Tumor treated with bacteria and non-immunogenic chemotherapy. Although host immunity is stimulated by bacteria, the cancer-specific immunity cannot be activated due to the lack of immunogenic cell deaths of cancer cells. (v) Tumor treated with bacteria and immunogenic chemotherapy. Both the innate and adaptive immune systems are activated due to the synergistic effects of DAMPs and PAMPs.

CD25 antibody (Biolegend; Cat# 102033; 1:50), APC CD80 antibody (BD Pharmingen Cat# 560016; 1:100), PE CD86 antibody (BD Pharmingen Cat# 553692; 1:100), PE IFN-γ antibody (Biolegend; Cat# 505808; 1:200), Brilliant Violet 785™ TNF-α antibody (Biolegend; Cat# 506341; 1:200).

### Statistical analyses

All statistical analyses were done using GraphPad Prism 8.0. Statistical differences among different treatment groups of all tumor-growth curve were computed using two-way ANOVA analysis (when groups >3) with Tukey's multiple comparisons test while paired t-test (two-tailed) was used when only two groups were required for comparison. On the other hand, statistical differences among column groups were computed using one-way ANOVA (when groups >3) with Tukey's multiple comparisons test whereas unpaired t-test (two-tailed) were applied when only two column groups were involved.

### Preparation of drug stocks for in vitro and in vivo assays

All drug stocks were freshly prepared in PBS (pH 7.2) prior to the in vitro or in vivo experiments and sonicated using a water bath sonicator to aid dissolution. For in vitro cytotoxicity assays, 5-FU and oxaliplatin stocks were prepared as a 5 mM stock and diluted with culture medium to the desire concentrations for subsequent

assays. For the in vivo assays, 5-FU and oxaliplatin were prepared as a 4 mg/mL and 1.25 mg/mL stock, respectively.

### Cytotoxicity assays of chemotherapeutics on MC38 cell line

MC38 cell line was maintained in DMEM medium supplemented with 10% FBS, 10 mM HEPES, 2 mM glutamine, 0.1 mM non-essential amino acids and 1 mM sodium pyruvate at 37 °C and 5% $CO_2$. For cytotoxicity assay, $5 \times 10^3$ cell/well of MC38 cells were seeded into a 96-well plate and allowed to stabilize overnight. The cells were then treated with various concentrations of 5-FU and oxaliplatin for 48 h. The resulting cell viability was determined using CCK-8 assay by replacing the old medium with 100 μL of fresh medium containing 10 μL of the CCK-8 reagent, followed by 1.5-h incubation at 37 °C. The resulting $A_{450}$ signal was measured using a Tecan-Infinite M1000 PRO microplate reader and $IC_{50}$ values were calculated using GraphPad Prism8 software. All experiments were performed in triplicate.

### Cytotoxicity assays of chemotherapeutics on *E. coli* (DH5α)

*E. coli* (DH5α) glycerol stock was first streaked on a LB agar plate and incubate overnight at 37 °C. A single colony was then picked and cultured in LB medium at 37 °C, overnight. The bacteria

culture was then diluted to 0.1 $OD_{600}$ (3 mL per sample) and incubate with different concentrations of either drug (5-FU or oxaliplatin) at 37 °C and 200 rpm shaking. The growth kinetics ($OD_{600}$) of E. coli (DH5α) were monitored hourly for up to 6 h and at a 24-h endpoint using a DEN-600 photometer (BioSan) and $IC_{50}$ values were calculated using GraphPad Prism8 software. All experiments were performed in triplicate.

## Starvation of MC38 cell line and *E. coli*

A zero-nutrient approach was used to assess the effects of starvation on cancer cells (MC38) and E. coli (DH5α). Specifically, $2 \times 10^4$ of MC38 cells were incubated in PBS for a predetermined period (0–20 h) and the post-starvation cell viability was measured using CCK-8 assay. Meanwhile, $4 \times 10^7$ of E. coli (DH5α) was incubated in PBS for 1 or 20 h prior to inoculation into LB medium. The growth kinetics of each group were monitored hourly by $OD_{600}$ measurement for up to 6-h post-starvation.

## In vivo anti-tumor assays

Wild-type C57BL/6 (B6) and Balb/c mice were obtained from the National Laboratory Animal Center (NLAC), NAR Labs, Taiwan. All animal study was conducted in a pathogen-free environment in accordance to the scientific and ethical guidelines approved by the Animal Care and Usage Committee of Academia Sinica (IACUC #18-09-1224). Mice were maintained at the animal house facilities of Institute of Biomedical Sciences, Academia Sinica with proper temperature (19–23 °C) and humidity (50–60%) control, under a 12-h light dark cycle. To induce tumor in mice, $5 \times 10^5$ of MC38/CT26 cells were suspended in PBS (pH 7.2) and subcutaneously injected into the right flank of each mouse. Treatment was initiated when the diameter of tumor was at least 3 mm. Tumor volume (mm³) was calculated using the following equation: [(tumor length + tumor weight)/2]³ × 0.52. Mice with a tumor that was larger than 1500 mm³ were considered dead for ethical reasons. The study was performed as an unblinded study.

## In vivo anti-tumor assays of 5-FU

The mice were grouped into four treatment group (n = 5) in the 5-FU treatment regime consisting of solvent (PBS), drug only (5-FU), bacteria only (E. coli, DH5α) and combination treatment (5-FU + E. coli, DH5α). Mice receiving 5-FU were given 40 mg/kg of 5-FU intraperitoneally (IP) thrice per week for a total of 7 doses, while mice receiving bacteria treatment were given intratumoral (IT) injection of $4 \times 10^8$ of E. coli (DH5α) in PBS using a 29-gauge insulin syringe on the first day of the treatment. The combination treatment group received both 5-FU (IP) and E. coli (DH5α; IT) treatments.

## In vivo anti-tumor assays of 5-FU with starvation

The mice were grouped into 3 treatment groups (n = 3): PBS, combination (5-FU + E. coli, DH5α) and combination with intermittent starvation. The procedure for PBS and combination groups was identical as the aforementioned treatment protocol. However, mice in the intermittent starvation group received an addition of two non-consecutive 24-h starvation per week with a total of 4 sessions throughout the treatment period.

## In vivo anti-tumor assays of oxaliplatin

The mice were divided into 4 groups (n = 5) under this treatment regime: PBS, drug only (oxaliplatin), bacteria only (E. coli, DH5α) and combination (oxaliplatin + E. coli, DH5α). Mice receiving oxaliplatin (O) were given 5 mg/kg of oxaliplatin intraperitoneally (IP) every 3 days for a total of 7 doses. Meanwhile, mice receiving bacteria treatment were given intratumoral (IT) injection of $4 \times 10^8$ of E. coli (DH5α) in PBS using a 29-gauge insulin syringe, on the 1st and 8th day of treatment. The combination treatment group received both the oxaliplatin (IP) and E. coli (DH5α; IT) treatments. The mice that had their tumors eradicated were re-challenged by subcutaneous inoculation of $2.5 \times 10^5$ MC38 cells on the collateral flank. The rechallenge was performed at least 3 weeks after the last administration of E. coli and two weeks after no tumor growth was observed in mice. A set of naïve mice (n = 3) were included as tumor growth control. Subsequently, an additional set of experiments were conducted to assess the feasibility of intravenous (IV) administration of E. coli (DH5α) into the subjects treated. The procedures were identical to the IT counterpart, except 4 doses of $1 \times 10^8$ E. coli (DH5α) were IV administered every three days through the tail veins of the mice.

## Tumor-infiltrating leukocytes (TILs) analysis

The tumors from mice in different treatment groups were excised, weighted, and processed to obtain tumor cell suspensions. Tumor growth curves of mice for TILs analysis are available as Appendix Fig. S7A,B. The tumor-harvesting time point is set at least 7 days post the last administration of E. coli due to our observations that innate immune cells, particularly neutrophils dominated the tumor within 72 h of bacteria inoculation while T cells required a longer time frame to be fully activated and infiltrates into the TME (Appendix Fig. S7C). The tumors were cut into small pieces, immersed in TILs buffer (composed of 670 μg/mL collagenase IV and 2 μg/mL DNase I dissolved in MC38 culture medium) and processed into single-cell homogenates using the gentleMACS™ Octo Dissociator (Miltenyi Biotec Inc). The resulting cell suspensions were passed through a 40 μm cell strainer, treated with 1X RBC lysis buffer, and resuspended in FACS buffer (5% FBS in PBS, v/v, pH 7.2). All samples were blocked with CD16/32 Fc blocker (1:50 dilution) prior to immunostaining. For surface staining, the TILs were stained with eFluor™780 eBioscience™-Fixable Viability Dye, Pacific Blue™-CD45 antibody, PerCP/Cyanine5.5-CD45 antibody, APC-CD4 antibody, AlexaFluor®-CD8a antibody, Alexa Fluor®647-Ly6G antibody, FITC-CD11B antibody, Pacific Blue-CD11B antibody, BrilliantViolet421-F4/80 antibody and FITC-MHCII antibody. For intracellular staining, all samples were fixed and permeabilized with 4% paraformaldehyde (PFA) and 0.1% saponin, respectively, after staining of the surface markers. The samples were then stained with PE-IFN-γ antibody and Brilliant Violet 785™-TNF-α antibody. The stained samples were analyzed using Attune NTX flow cytometer, and the data were processed using FlowJo V10 software.

## Immune cells/cancer cells/bacteria co-culture assay

Spleenocytes were isolated by grinding the spleen of murine through a 40 μm strainer followed by RBC lysis and cell counting.

**The paper explained**

**Problem**

Solid tumors, functioning as ecosystems conducive to tumor cell growth, typically establish tumor cells as the dominant species with limited competition within the tumor microenvironment (TME). These tumor cells acquire superior fitness through genomic instability and high proliferation via microevolutionary processes, concurrently influencing immune cells to adopt a pro-tumoral phenotype. Anticancer therapies often fail to disrupt tumor cell dominance, thus leading to tumor relapse.

**Results**

We combined bacterial therapy with various treatment modalities to establish a Differential Selective Pressure (DSP) that negatively impacts cancer cell survival, and treatments that favor bacterial survival over tumors were found to enhance bacterial therapy efficacy. Notably, the combination of oxaliplatin with *E. coli* completely eradicated tumors in a syngeneic mouse colorectal cancer model. Analysis of tumor-infiltrating and cultured immune cells revealed that this co-treatment stimulated both innate and adaptive immune responses.

**Impact**

Our study supports that introducing a competing species into the TME along with manipulating the DSP can synergistically suppress and eradicate tumors. A key factor for optimal anti-tumor activity is the alteration of the immune status within the TME through an immunogenic chemotherapeutic (oxaliplatin). This underscores the importance of a thoughtful combination choice and unveils a broad combinatorial space for further exploration with different chemotherapeutic agents and bacteria in diverse cancer contexts.

The spleenocytes were maintained at 37 °C in RPMI medium supplemented with 10% heat-inactivated FBS, 0.05 mM β-mercaptoethanol, 10 mM HEPES, 2 mM glutamine, 0.1 mM non-essential amino acids, 1 mM sodium pyruvate, 1X insulin-transferrin-selenium (ITS), 100 U/mL human IL-2 and 2 ng/mL mouse IL-7. For activation assay, $2 \times 10^6$ freshly isolated spleenocytes were seeded individually or in combination with 2 μM oxaliplatin, $2 \times 10^6$ *E. coli* (DH5α) or $2 \times 10^5$ MC38 in a 12-well tissue culture plate. It should be noted that the MC38 cells in the groups receiving oxaliplatin were pretreated with 2 μM of oxaliplatin for 24 h prior to the assay. The composition of immune cell types was determined using flow cytometry at 24 h post-treatment. In addition, a parallel experiment was conducted using MC38-EGFP cell line under identical conditions. The cells were lifted using Versene (Gibco™), pelleted and stained using eFluor™ 780 eBioscience™ Fixable Viability Dye prior to flow cytometry analysis.

**Assessing cytotoxicity of oxaliplatin (CCK-8 assay) toward MC38 in the presence of spleenocytes and *E. coli***

The cytotoxicity assay was performed under conditions similar to those described in the aforementioned procedure. To assess oxaliplatin cytotoxicity in the presence of *E. coli* or spleenocytes, either $5 \times 10^4$ of *E. coli* or spleenocytes were co-incubated with MC38 ($5 \times 10^3$ cell/well) in the presence of 0–50 μM of oxaliplatin for 48 h, followed by determination of cell viability using CCK8 assay. To minimize interference from spleenocytes and *E. coli* on the CCK-8 assay results, which primarily reflect the count of MC38

cells, the cultures were gently washed twice with warm PBS to remove non-adherent cells. Additionally, a preliminary assay was performed to assess the impact of live spleenocytes on the CCK-8 assay (Appendix Fig. S13B,C). Importantly, MC38 was cultured in spleenocytes culture medium in all assays requiring co-incubation with spleenocytes. For visualization of spleenocytes adherence toward MC38 cells, fresh murine spleenocytes were pre-labeled with 10 μM of CFSE ($1 \times 10^6$ of spleenocytes/mL of CFSE in PBS) for 20 min at 37 °C. The labeled cells were washed thrice with warm PBS followed by co-incubation with MC38, oxaliplatin or *E. coli* for 48 h. All samples were gently washed twice with 100 μL of warm PBS prior to the observation to remove non-adherent spleenocytes. Observations were done using Nikon Eclipse Ti2 fluorescence microscope. The size of CFSE-labeled spleenocyte clusters were determined using the analysis software, NIS elements Br 5.30.03 and plotted using GraphPad Prism8.

## Data availability

This study includes no data deposited in external repositories.

## Peer review information

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

## Acknowledgements

The work was supported by the National Science and Technology Council of the Republic of China (NSTC-110-2113-M-001-064-MY3) and Academia Sinica, Taiwan (AS-CDA-108-L07).

## Author contributions

**See-Khai Lim**: Conceptualization; Data curation; Formal analysis; Investigation; Methodology; Writing—original draft; Writing—review and editing. **Wen-Ching Lin**: Data curation; Formal analysis; Investigation. **Sin-Wei Huang**: Data curation. **Yi-Chung Pan**: Data curation; Formal analysis. **Che-Wei Hu**: Data

curation; Formal analysis. **Chung-Yuan Mou**: Data curation; Supervision; Writing—review and editing. **Che-Ming Jack Hu**: Formal analysis; Supervision; Investigation; Writing—review and editing. **Kurt Yun Mou**: Conceptualization; Resources; Formal analysis; Supervision; Funding acquisition; Validation; Methodology; Writing—original draft; Writing—review and editing.

## Disclosure and competing interests statement

The authors declare no competing interests.

