## [Peer Review File · EMBO Molecular Medicine]

Bacteria colonization within tumors creates a favorable niche for immunogenic chemotherapy

See-Khai Lim, Wen-Ching Lin, Sin-Wei Huang, Yi-Chung Pan, Che-Wei Hu, Chung-Yuan Mou, Che-Ming Hu, and Kurt Yun Mou
DOI: [10.15252/emmm.202318067](https://doi.org/10.15252/emmm.202318067)

Corresponding author: Che-Ming Hu (chu@ibms.sinica.edu.tw)

Review Timeline:

Submission Date:	30th May 23
Editorial Decision:	13th Jul 23
Revision Received:	2nd Nov 23
Editorial Decision:	28th Nov 23
Revision Received:	18th Dec 23
Accepted:	21st Dec 23

Editor: *Zeljko Durdevic*

Transaction Report:

13th Jul 2023

Dear Dr. Mou,

Thank you for the submission of your manuscript to EMBO Molecular Medicine, and please accept my apologies for the delay in getting back to you. We have now received feedback from two of the three reviewers who agreed to evaluate your manuscript. As the referee #2 will unfortunately not be able to return his/her report in a timely manner, and given that both reviewers provide very similar recommendations, we prefer to make a decision now in order to avoid further delay in the process.

As you will see from their reports pasted below, both referees recognize potential interest of the study but also raise serious and partially overlapping concerns that should be addressed in a major revision. Particularly, in vitro co-culture experiments need to be significantly improved to support the conclusions on the mechanism. Should referee #2 provide a report, we will send it to you, with the understanding that we will not ask for an additional revision. If you would like to discuss further the points raised by the referees, I am available to do so via email or video. Let me know if you are interested in this option.

We would welcome the submission of a revised version within three months for further consideration. Please let us know if you require longer to complete the revision.

I look forward to receiving your revised manuscript.

Yours sincerely,

Zeljko Durdevic

We require:

- 1) A .docx formatted version of the manuscript text (including legends for main figures, EV figures and tables). Please make sure that the changes are highlighted to be clearly visible.
- 2) Individual production quality figure files as .eps, .tif, .jpg (one file per figure). For guidance, download the 'Figure Guide PDF': (<https://www.embopress.org/page/journal/17574684/authorguide#figureformat>).
- 3) A .docx formatted letter INCLUDING the reviewers' reports and your detailed point-by-point responses to their comments. As part of the EMBO Press transparent editorial process, the point-by-point response is part of the Review Process File (RPF), which will be published alongside your paper.
- 4) A complete author checklist, which you can download from our author guidelines (<https://www.embopress.org/page/journal/17574684/authorguide#submissionofrevisions>). Please insert information in the checklist that is also reflected in the manuscript. The completed author checklist will also be part of the RPF.

6) It is mandatory to include a 'Data Availability' section after the Materials and Methods. Before submitting your revision, primary datasets produced in this study need to be deposited in an appropriate public database, and the accession numbers and database listed under 'Data Availability'. Please remember to provide a reviewer password if the datasets are not yet public (see <https://www.embopress.org/page/journal/17574684/authorguide#dataavailability>).

13) Author contributions: You will be asked to provide CRediT (Contributor Role Taxonomy) terms in the submission system. These replace a narrative author contribution section in the manuscript.

14) A Conflict of Interest statement should be provided in the main text.

15) Every published paper now includes a 'Synopsis' to further enhance discoverability. Synopses are displayed on the journal webpage and are freely accessible to all readers. They include a short stand first (maximum of 300 characters, including space) as well as 2-5 one-sentence bullet points that summarize the paper. Please write the bullet points to summarize the key NEW findings. They should be designed to be complementary to the abstract - i.e. not repeat the same text. We encourage inclusion of key acronyms and quantitative information (maximum of 30 words / bullet point). Please use the passive voice. Please attach these in a separate file or send them by email, we will incorporate them accordingly.

Please note: When submitting your revision you will be prompted to enter your funding and payment information. This will allow Wiley to send you a quote for the article processing charge (APC) in case of acceptance. This quote takes into account any reduction or fee waivers that you may be eligible for. Authors do not need to pay any fees before their manuscript is accepted and transferred to the publisher.

EMBO Press participates in many Publish and Read agreements that allow authors to publish Open Access with reduced/no publication charges. Check your eligibility: <https://authorservices.wiley.com/author-resources/Journal-Authors/open-access/affiliation-policies-payments/index.html>

***** Reviewer's comments *****

Referee #1 (Remarks for Author):

This is a well-developed and -written study by Lim et. al describing a synergistic anticancer effect of combined treatment of syngeneic murine colon cancer models with the immunogenic chemotherapeutic oxaliplatin and primarily intratumoral application of E. coli bacteria. Both in vitro co-culture and in vivo experiments support this hypothesis based on a co-stimulatory effect derived from DAMP and PAMP signals. The data appear valid and the experiments performed in a sound manner.

Points for improvement:

The manuscript at the beginning is focused on the competition effect and therapies differentially disfavoring the human cancer cells over the bacteria. This concept of differential stress resistance (DSR) dominates the abstract and introduction, but is more or less lost in the second part of the manuscript, describing straight-forward the synergism in activation of innate and adaptive immune components. The discussion section only randomly mentions the DSR and competition concept. Such, the argumentation line of the manuscript needs some harmonization.

The "fasting" experiment needs to be better embedded into the manuscript. It appears somehow disconnected!

Concerning the translational aspects of the in vivo experiments, side-effects of the E. coli therapy need to be more closely monitored. What is the impact, especially of the intravenous bacteria application, on the immune conditions in the peripheral blood? What happens with systemic cytokine levels etc.? Is there any septic reaction and if not, please explain why?

The immune cell analyses in Figures 3 and 4 have obviously been done from the tumors at the end of the in vivo experiment, as exemplarily shown in Figure 2i. Does it make any sense to compare the immune compartments at this very late stage of therapy with minimal tumor volume especially in the combined treatment group. An experiment starting therapy with a more progressed tumor volume and then analyzing the tumor immune invasion during the response phase should be included!

The in vitro co-culture experiment is elegant but the inclusion of pen/strep to avoid E. coli overgrowth is kind of problematic. It has to be stressed that you see in these experiments primarily the effect of dead bacteria, which should be different in the in vivo setting.

Minor points:

Legend of the color codes in Figure 4e are missing!

Statistics in Figure 5a are missing and generally a chapter on statistics would be desirable.

How can tumor-specific viability be seen in the CCK8 assay when viable splenocytes (and probably even some living bacteria) are present? Please explain!

Is Figure S11 mentioned in the results text? Maybe I missed it because many suppl. Figures are just mentioned randomly and as a bundle (e.g. S12- S15).

Referee #3 (Remarks for Author):

In their manuscript, Lim et al. present data on bacteria mediated cancer therapy. They use murine transplantable colon carcinoma (mainly M38 in C57Bl/6 mice), a laboratory strain of *E. coli* and co-treat the tumor bearing mice with 5-FU or oxaliplatin. The combined treatment of the cancer bearing mice with *E. coli* and oxaliplatin shows in some cases complete clearance of the tumors. The authors present then data on tumor infiltration by cells of the innate immune system and their differentiation status as well as the infiltration by cells of the adaptive immune system. This part of the manuscript is highly interesting and the reason for my positive evaluation of the manuscript although some weak point need to be ruled out first. The authors then try to show the mechanism of the effect of co-administration by in vitro co-culture. This part is too preliminary and not convincing at all. It is completely over interpreted and technically extremely weak. It should be deleted from the manuscript and worked out in more details and more convincingly. In details:

1. Results: it should be pointed out that the administration of the bacteria was done intratumorally. How often the bacteria were applied should also be mentioned when appropriate.
 2. Results: the Kaplan Meier blot in Fig. 1h shows no difference between 5-FU +E and 5-FU+E+starvation. In the text a difference is claimed.
 3. Results: why was starvation not tested for Oxa+E. coli?
 4. Results: Fig. 2l. The same spleen size is claimed for O and OE but 59mg compared to 89mg is for me not the same.
 5. Results: in general, all flow cytometry data are displayed as percentage. In some cases, this might still be meaningful but only in combination with absolute numbers all these data are acceptable.
 6. Results: although OXA depletes all tumor infiltrating immune cells, which the authors blaim on the cytotoxicity of Oxa. Why do those cells survive when *E. coli* are co-administered? This needs to be explained either here or in the discussion.
 7. Results: Fig. 4a-d. Only TNF-a and IFN-g is used as activation marker. Additional markers including other activation markers need to be included in the analysis.
 8. Results: Fig. 5e. Legend is missing. It is completely unclear what is shown here.
 9. Results in vitro: as pointed out already, these data are extremely preliminary. Fig. 5. These data are meaningless unless the absolute numbers are shown. Numbers of innate immune cells like DC or macrophages increase. Are they proliferating? This is unlikely since they are considered to be end cells. Under some conditions, they might proliferate. This needs to be shown. Why was isolated LPS not tested?
 10. Results: the authors then investigate the reactivity of spleen cells from naïve mice towards the tumor cells. First of all, it is not clear how they determine cytotoxicity or inhibition in the mixture of tumor and spleen cells. Then they claim that the spleen cells specifically attack the tumor cells when Oxa and *E. coli* are in the co-culture. They try to show this by pictures of aggregates between tumor cells and spleen cells after two days of co-culture without staining the T cells. Do the authors really believe that that many tumor specific T cells are in the spleen of a naïve mouse?
 11. Discussion: it is not true that bacteria do not support the adaptive immune response against tumors. This has been shown in several publications.
 12. Supplement: in general, all the flow cytometry panels are too small. The writing here is most of the time unreadable.
- Minor points: the English of the manuscript is well readable. Only at a few locations the wording is miss-understandable.
- Results: 7. Line: "than that for *E. coli*" (not of). Page 8, last line: "monotreatment" not single treatment since the bacteria have been injected several times.

***** Reviewer's comments *****

Referee #1 (Remarks for Author):

This is a well-developed and -written study by Lim et. al describing a synergistic anticancer effect of combined treatment of syngeneic murine colon cancer models with the immunogenic chemotherapeutic oxaliplatin and primarily intratumoral application of *E. coli* bacteria. Both in vitro co-culture and in vivo experiments support this hypothesis based on a co-stimulatory effect derived from DAMP and PAMP signals. The data appear valid and the experiments performed in a sound manner.

Points for improvement:

1. The manuscript at the beginning is focused on the competition effect and therapies differentially disfavoring the human cancer cells over the bacteria. This concept of differential stress resistance (DSR) dominates the abstract and introduction, but is more or less lost in the second part of the manuscript, describing straight-forward the synergism in activation of innate and adaptive immune components. The discussion section only randomly mentions the DSR and competition concept. Such, the argumentation line of the manuscript needs some harmonization.

Response: We extend our gratitude to the reviewer for acknowledging the manuscript's quality and its relevance to the field of cancer therapy. Having meticulously reviewed our work, we acknowledged the disconnect between the concept of DSR and the later part that focused on tumor immune environment. As such, modification to the text and several display items have been made to harmonize the article. Details are as follows:

- In the framework of DSR, we have revised the abstract and introduction to connect the immune status within the TME as an integral aspect of the DSR theory. In addition, we updated Figure 7 to highlight differential selective pressure in the TME in response to different combinatorial treatment strategies. Several transition paragraphs are also provided in the body and discussion section to relate immune cell modulation to DSR principle. These transition paragraphs are as described below.
- *“We hypothesized that achieving long-term tumor remission often relies on the engagement of immune responses, in particular the host adaptive immunity. However, within the tumor microenvironment (TME), tumor-infiltrating leukocytes tend to adopt a pro-tumoral phenotype. While introducing *E. coli* into the TME can activate a fraction of the immune system, the resulting selective pressure is non-specific and primarily targeting the bacteria rather than the tumor cells. To align with our concept of establishing DSR within the TME to target tumor cells, we then decided to explore an immunogenic drug with the ability to direct the host immune responses towards tumor cells.”*
- **Discussion section:** *“Solid tumors can be seen as ecosystems where the microenvironment is optimized for tumor cell growth. In ecological terms, tumor cells are the dominant species with limited competition from other members within the TME. In attempting to break this dominance, we introduced an invasive species (*E. coli*) into the TME to compete for survival resources. However, our experiment showed that bacteria monotreatment, although temporarily controlled tumor growth, could not achieve tumor eradication. Combinatorial treatments, such as 5-FU and fasting, were designed to selectively favor bacteria over cancer cells. While these combinatorial treatments enhanced tumor suppression, complete tumor remission remained elusive. Consequently, we modified the survival and selective factors in the TME by introducing an immunogenic drug, oxaliplatin, which in combination with bacterial therapy converts the immunosuppressive TME to an immunoreactive one [43, 44].”*

2. The "fasting" experiment needs to be better embedded into the manuscript. It appears somehow disconnected!

Response: We thank the reviewer for this suggestion. We have substantially revised the abstract, introduction, and Figure 7 to provide better justification for the “fasting” experiment. In short, we show that nutrient deprivation induces DSR that favors bacteria growth over cancer, and the combination treatment results in improved tumor suppression. We highlight that the anticancer enhancement due to fasting *in vivo* was modest due to practical limitations in executing nutrient deprivation in animals, and to facilitate transition of the study into the latter half of the section, we included additional paragraphs to justify design rationale towards combining bacterial therapy with immunogenic chemotherapy. The added paragraphs on page 7 are as follows:

“Although the DSR provided by 5-FU, fasting, and E. coli significantly enhanced tumor suppression, the treatments were not able to achieve complete tumor remission. We hypothesized that achieving long-term tumor remission often relies on the engagement of immune responses, in particular the host adaptive immunity.”

3. Concerning the translational aspects of the *in vivo* experiments, side-effects of the *E. coli* therapy need to be more closely monitored. What is the impact, especially of the intravenous bacteria application, on the immune conditions in the peripheral blood? What happens with systemic cytokine levels etc.? Is there any septic reaction and if not, please explain why?

Response: We thank the reviewer for the comment as safety stands as a paramount concern in any form of bacterial cancer therapy.

- Aligning with this principle, we selected a non-pathogenic *E. coli* strain (DH5 α) as the bacteria treatment. The safety profiles of DH5 α have been subject to comprehensive investigation [1]. In addition, the safety of bacteria cancer therapy of non-pathogenic bacteria (such as *E. coli* DH5 α /Nissile 1917) had been thoroughly assessed in mice model including survivability in blood, organs [2], serum cytokine profiles [3, 4] and organ histology [5]. All the safety assessments showed that the non-pathogenic strains of *E. coli* are safe to be administered intravenously.
- Furthermore, we assessed the safety profile of the bacteria by closely monitoring weight fluctuations and blood biochemical parameters in mice who received intravenous DH5 α treatment. Mice received four-IV doses (1×10^8 *E. coli*/dose) showed no significant decrease in their body weight, indicating the dose applied is tolerable (Figure R1A; included in Figure S5B in the manuscript’s supplementary information).
- For whole blood cells analysis. We observed a steep increase of neutrophils (43.56 %) at day 1 post IV inoculations of *E. coli* in comparison to the untreated control (26.2 %), indicating the first line immune defense against bacteria was activated. The effects persist till day 3 post inoculation and the readings fall back into the normal range at day-7 post inoculation (23.04 %) (Figure R1C; included as Figure S5E in the manuscript’s supplementary information).
- In addition, we also conducted a hemolysis assay using mice RBC and assessed the effects of hemolysis by reading the A570 value of samples’ supernatant for quantification of the release of hemoglobin into the solution after 30-min of co-incubation with *E. coli* (Figure R1B; included as Figure S5D in the manuscript’s supplementary information).

Figure R1. Safety assessment of systemic *E. coli* administration in mice models. (A) Body weight changes (percentage relative to the same mice prior *E. coli* IV administration) of mice received *E. coli* treatment. A total of 4 doses (1×10^8 bacteria in 100 μ L of PBS/dose) were given every 3 days throughout the treatment period. (B) Assessment of hemolytic potential of *E. coli* using murine RBC. Murine blood was extracted via cardiac blood collection. The blood was kept in EDTA tube followed by RBC isolation (900 xg centrifugation for 5 min and wash thrice with sterile PBS). The hemolysis assay was performed by incubating 4×10^7 *E. coli* with 1 mL of 1% murine RBC at 37 $^{\circ}$ C for 30 min. The resulting samples were then centrifuged at 900 xg for 5 min and hemolysis (release of hemoglobin) was determined by reading the absorbance of the at 570 nm. A negative control (treated with PBS) and a positive control (1% triton-X 100) were included. (C) Whole blood cell analysis of mice received 1-dose of IV-administrated *E. coli* (4×10^8 bacteria/dose). Blood samples were collected via cheek pouch method at Day-1, Day-3 and Day-7 post-treatment.

- In addition, in our co-culture study, we observed that DH5 α can only proliferate normally in RPMI medium when heat-inactivated FBS was used, possibly due to the presence of complement system that could bind to the bacteria and lead to lysis of the bacteria. This observation was also confirmed by other independent studies: “*In contrast, strains EQ1, DH5 α , BLR and BL21 were only able to survive in sera which had been heat inactivated and the numbers of viable bacteria were reduced to levels below detection by 30 min.*” [1].
4. The immune cell analyses in Figures 3 and 4 have obviously been done from the tumors at the end of the in vivo experiment, as exemplarily shown in Figure 2i. Does it make any sense to compare the immune compartments at this very late stage of therapy with minimal tumor volume especially in the combined treatment group. An experiment starting therapy with a more progressed tumor volume and then analyzing the tumor immune invasion during the response phase should be included.

- **Response :** We express our gratitude to the reviewer for raising this question. Our observations indicate that, at earlier time points (D1 and D3 post *E. coli* injection; tumor volumes were approximately 300-500 mm³ at point of analyses), neutrophils predominate among the tumor-infiltrating immune cells, as illustrated in the figure below. It is worth noting that innate immunity typically responds rapidly, whereas adaptive immunity typically requires more time to reach its peak, with a lag of approximately 5-7 days post-exposure [6, 7].
- Given this understanding, we deliberately selected a later time point to assess the activation and infiltration status of T lymphocytes within the tumor. However, we also acknowledge that the infiltration patterns of immune cells at earlier time points provide valuable additional insights for our readers. Consequently, we are pleased to incorporate this data into the extended view (EV) section of the manuscript for the benefit of our audience (Figure R2; included as Figure S7C in the manuscript's supplementary information).

Figure R2. Composition of the tumor infiltrating immune cells in each treatment group. For mice received *E. coli* treatment (IT), three end points were applied for tumor harvesting and immune cells profiling (Day 1, 3 and 8 after the last dosage of *E. coli* administration).

5. The *in vitro* co-culture experiment is elegant but the inclusion of pen/strep to avoid *E. coli* overgrowth is kind of problematic. It has to be stressed that you see in these experiments primarily the effect of dead bacteria, which should be different in the *in vivo* setting.

Response:

- We appreciate the reviewer's positive feedback, and in response, we would like to present some preliminary data to provide a rationale for our choice of using PenStrep (dead bacteria) instead of live bacteria in the *in vitro* experiments. We performed some additional studies to compare the effects of living and dead bacteria towards splenocytes in an identical co-culture system.
- Due to the strong proliferative ability of *E. coli*, we inoculated 10 X fewer *E. coli* (2×10^5 cells for live *E. coli*) and shortened the incubation period from 48 h to 24 h (MOE_L). However, even under these modified conditions, we still observed overgrowth of live *E. coli*, characterized by slightly turbid medium and a color shift from red to orange. Subsequently, we performed flow cytometry

analysis to assess the composition of spleenocytes that were co-incubated with MC38, *E. coli*, and oxaliplatin.

- Interestingly, we noted a higher proportion of dead spleenocytes compared to their PenStrep-treated counterparts. This difference can be attributed to the intense nutrient competition among the three components in the culture medium. However, despite these viability disparities, our analysis of immune cell activation patterns revealed consistent findings. Both the groups with live or dead *E. coli* exhibited (as Figure R3; included as Figure S12 in the manuscript's supplementary information):
 1. Upregulations of the CD25 marker in CD4+ and CD8+ T-cells.
 2. Increased populations of CD80+/CD86+ dendritic cells and macrophages.
 3. Enhanced MHCII expression on all antigen-presenting cells.
- Based on these observations, we rationalize that the addition of PenStrep would not alter the main conclusions of our *in vitro* spleenocyte assays. Additionally, it helps to mitigate the complications introduced by bacterial overgrowth and nutrient competition, ensuring the reliability and consistency of our *in vitro* assay outcomes.

Figure R3. Comparison of the effects of *in vitro* spleenocytes activation using live (no PS; MOE_L) versus growth-inhibited bacteria (with PS; MOE). (A) Percentage of dead spleenocytes 24 hours post-treatment. (B) Percentage of activated T-cells (CD25+) 24 hours post-treatment. (C) Percentage of CD80+/86+ population for each type of APCs at 24 hours post-treatment. (D) Percentage of MHCII+ population for each type of APCs at 24 hours post-treatment.

6. Legend of the color codes in Figure 4e are missing!

Response: We thank the reviewer for bringing this matter to our attention. We have now reintroduced the color codes to the figure as per your suggestion.

7. Statistics in Figure 5a are missing and generally a chapter on statistics would be desirable.

Response: We extend our gratitude to the reviewer for addressing this concern. In response, we have re-executed the experiment and reconstructed Figure 5 to display the absolute counts of each spleenocyte population. This modification serves to provide a more accurate representation of the proliferation of each spleenocyte population in our study, in line with the professional suggestion made by the other reviewer. Additionally, we have incorporated a paragraph in the methodology section that delineates the statistical analysis parameters employed in this study for clarity and transparency.

“All statistical analyses were done using GraphPad Prism 8.0. Statistical differences among different treatment groups of all tumor-growth curve were computed using 2-way ANOVA analysis (when groups > 3) with Tukey's multiple comparisons test while paired t-test (two-tailed) was used to compute statistical difference between two growth curves (group = 2). On the other hand, statistical differences between different column groups were computed using 1-way ANOVA (when groups > 3) with Tukey's multiple comparisons test whereas unpaired t-test (2-tailed) were applied for comparison between two column groups.”

8. How can tumor-specific viability be seen in the CCK8 assay when viable splenocytes (and probably even some living bacteria) are present? Please explain!

Response: We appreciate the reviewer's query, and we would like to offer clarification on this matter by presenting some of our preliminary data obtained during assay optimization.

- Before conducting the CCK8 assay, we performed two washes with PBS in each well. This step was taken because most bacteria and spleenocytes are non-adherent, except for those adhered to the surface of MC38.
- Furthermore, it is important to note that signal changes in CCK8 assays primarily depend on the metabolic activity of the cells (conversion of WST-8 dye into colored formazan by cellular dehydrogenases) under investigation. In preparation for the assays, we carried out an analysis using spleenocytes alone to assess their reactivity to the CCK8 reagent. This analysis revealed that the metabolic activity of spleenocytes was considerably slower compared to that of MC38 cells. In Figure R4A (included as Figure S13C in the manuscript's supplementary information), we have included a standard curve depicting the relationship between the number of spleenocytes and the A450 signal derived from the CCK8 assay.
- Additionally, we presented the normalized absorbance values from the co-culture assays, assuming an A450 value equivalent to 5×10^4 spleenocytes (assuming no spleenocytes were washed away). It is evident that the contribution of spleenocytes to the A450 signal was minimal, and as such, it should not significantly impact the results of the CCK8 assay. (as Figure R4; included as Figure S13B and S13C in the manuscript's supplementary information):

Figure R4. Assessment of CCK-8 signals for murine splenocytes. (A) Standard curve demonstrating correlation between splenocyte counts with the CCK-8 signals (A450). Splenocytes were counted using hemocytometer under microscope prior to the generation of standard curve. (B) Raw absorbance data for the MC38-splenocyte co-culture assay. In the co-culture assay, 5×10^4 splenocytes were added to each well and the corresponding CCK-8 signals were plotted on the graph with assumption of none of the splenocytes were washed away during the washing steps.

9. Is Figure S11 mentioned in the results text? Maybe I missed it because many suppl. Figures are just mentioned randomly and as a bundle (e.g. S12- S15).

Response: We are grateful to the reviewer for pointing out this matter, and we sincerely apologize for the omission in the manuscript body. The supplementary figures resulting from this concern have been appropriately described and integrated into the manuscript as follows:

- “However, an additional assay comparing the number of tumor-harboring *E. coli* showed no significant differences between mice that received only *E. coli* or *E. coli* and oxaliplatin, indicating the *in vivo* dosage applied in our study did not affect the survival and colonizing ability of *E. coli* (Fig S11a and b).”

Referee #3 (Remarks for Author):

In their manuscript, Lim et al. present data on bacteria mediated cancer therapy. They use murine transplantable colon carcinoma (mainly M38 in C57Bl/6 mice), a laboratory strain of *E. coli* and co-treat the tumor bearing mice with 5-FU or oxaliplatin. The combined treatment of the cancer bearing mice with *E. coli* and oxaliplatin shows in some cases complete clearance of the tumors. The authors present then data on tumor infiltration by cells of the innate immune system and their differentiation status as well as the infiltration by cells of the adaptive immune system. This part of the manuscript is highly interesting and the reason for my positive evaluation of the manuscript although some weak point need to be ruled out first. The authors then try to show the mechanism of the effect of co-administration by in vitro co-culture. This part is too preliminary and not convincing at all. It is completely over interpreted and technically extremely weak. It should be deleted from the manuscript and worked out in more details and more convincingly. In details:

1. Results: it should be pointed out that the administration of the bacteria was done intratumorally. How often the bacteria were applied should also be mentioned when appropriate.

Response: We thanked the reviewer for raising this issue, we had added in the administration route of *E. coli* and the chemotherapeutics in the Figure legend 1(c) and 2(c).

- “1(c) *In vivo anti-tumor activity of PBS, 5-FU, E. coli, and 5-FU+E. coli in the MC38 syngeneic murine model, 5-FU and E. coli were delivered intraperitoneally and intratumorally respectively.*” “2(c) *In vivo anti-tumor activity of PBS, OXA, E. coli and OXA+E. coli in the MC38 syngeneic murine mode, OXA and E. coli were delivered intraperitoneally and intratumorally respectively.*”
- We also stated the dosing of bacteria in the methodology section under the title “*In vivo Anti-tumor Assays of 5-FU*” and “*In vivo Anti-tumor Assays of Oxaliplatin*”.

2. Results: the Kaplan Meier blot in Fig. 1h shows no difference between 5-FU +E and 5-FU+E+starvation. In the text a difference is claimed.

Response: We thanked the reviewer for raising this issue and we apologize for the confusion of the graph. We had remade the Kaplan Meier plot to provide better clarity of the study. The figure is included as Figure 1H in the revised manuscript.

Figure R1. Kaplan Meier plot remade for Figure 1 (D) in the manuscript body.

- At the conclusion of this study, two-third of the mice in the starvation + 5FU + E treatment group survived the assay using tumor volume = 1500 mm³ as a cutoff point. In contrast, all mice in the solvent group and 5FU+E were determined to reach the endpoint of the study at day 14 and 16 respectively.

3. Results: why was starvation not tested for Oxa+E. coli?

Response: We express our gratitude to the reviewer for bringing up this question, and we recognize our previous manuscript did not properly justify the inclusion of starvation and its subsequent exclusion in the Oxa+E. coli treatment. To place the starvation study in its proper context, we have substantially modified the abstract, introduction, background, discussion, and Figure 7 to harmonize the different segments of the study. In short, inclusion of nutrient deprivation was designed to highlight strategies to enhance survival advantage of bacteria over cancer in different treatment combinations. In short, we show that nutrient deprivation induces DSR that favors bacteria growth over cancer, and the combination treatment results in improved tumor suppression. We point out that the anticancer enhancement due to fasting *in vivo* was modest due to practical limitations in executing nutrient deprivation in animals, and to facilitate transition of the study into the latter half of the section, we included additional paragraphs to justify design rationale towards combining bacterial therapy with immunogenic chemotherapy. The added paragraphs on page 7 are as follows:

“Although the DSR provided by 5-FU, fasting, and E. coli significantly enhanced tumor suppression, the treatments were not able to achieve complete tumor remission. We hypothesized that achieving long-term tumor remission often relies on the engagement of immune responses, in particular the host adaptive immunity.”

Furthermore, it's worth noting that the combination of oxaliplatin and intratumorally delivered *E. coli* has demonstrated the capability to completely eliminate tumors in our mouse model. Consequently, we decided to discontinue the starvation assay for the oxaliplatin-based treatment studies for both narrative and ethical reasons. We believe the revised manuscript better justifies our study design and appreciate the reviewer's suggestion.

4. Results: Fig. 2I. The same spleen size is claimed for O and OE but 59mg compared to 89mg is for me not the same.

Response: We express our gratitude to the reviewer for highlighting this issue and allowing us to clarify it. Our primary intent is to highlight that the combination of oxaplatin and *E. coli* reduced spleen enlargement, and we did not mean to claim that O and OE groups had the same spleen size. As such, we have revised the description text in the manuscript body to better describe this observation: *“Surprisingly, the OXA+E. coli co-treatment rescued the spleen size to a normal range in all tested mice (N=5), suggesting a reduction bacteria-associated systemic reactogenicity by the OXA combination.”*

Figure R2. Spleen weight of each treatment group at Day 15 post-treatment (Figure 2G; n = 5 for each treatment group and n = 4 for healthy control).

5. Results: in general, all flow cytometry data are displayed as percentage. In some cases, this might still be meaningful but only in combination with absolute numbers all these data are acceptable.
 - **Response:** We appreciate the valuable suggestions from the reviewer. Upon careful consideration of our data, we concur that utilizing absolute counts to analyze the flow cytometry data for *in vitro* assessment of spleenocyte composition provides a more direct and informative perspective regarding the stimulation (proliferation) or elimination of spleenocytes by each treatment. Consequently, we have taken the decision to repeat the experiment and present the results in the form of absolute counts, which we will address in subsequent points.
 - Regarding the *in vivo* analysis of Tumor-Infiltrating Lymphocytes (TILs), we understand and share your concerns and therefore, we included the flow cytometry data in the form of cell count in our supplementary data (S11). We maintained the data presentation in percentages in the manuscript body as the initial tumor volumes processed for TILs analysis can exhibit significant variation, especially when comparing combinations (OE) and the control (PBS) treatments, where the difference in tumor volume could exceed tenfold (e.g., 42 mg vs. 1820 mg). In our view, presenting the flow cytometry data in the form of percentages for the *in vivo* assays serves as a normalization method that helps account for differences in the initial sample sizes being processed. This approach helps mitigate the impact of varying tumor volumes on the analysis and allows for a more accurate assessment of treatment effects on TILs.
6. Results: although OXA depletes all tumor infiltrating immune cells, the authors blame on the cytotoxicity of Oxa. Why do those cells survive when E. coli are co-administered? This needs to be explained either here or in the discussion.
 - We appreciate the reviewer's question, and we have included in the discussion our hypotheses to explain the observed scenarios:
 - **Immunostimulant Compensation:** The addition of *E. coli*, a potent immunostimulant, may stimulate the proliferation of immune cells, thus compensating for the loss of immune cells due to the cytotoxic effects of chemotherapy. This hypothesis is supported by data from our *in vitro* spleenocyte analysis. Among all the cell types analyzed, antigen-presenting cells (APCs), including B cells, dendritic cells (DC), and macrophages, were relatively more sensitive to the cytotoxic effects of oxaliplatin. For example, the count of B cells decreased from 44,050/mL (in the M group) to 29,418/mL after 24

hours of incubation with 2 μ M of oxaliplatin. However, the addition of *E. coli* partially mitigated these effects, resulting in a B cell count of 39,424/mL in the MOE group (Figure R3; included as Figure 5A in the manuscript's body). Similar trends were observed in the DC and macrophage subsets. This hypothesis suggests that *E. coli* acts as a supportive factor in maintaining and stimulating certain immune cell populations, counteracting the cytotoxicity induced by oxaliplatin. Further analysis may be necessary to explore the mechanisms underlying this compensatory effect and its relevance to the observed outcomes in our study.

- **Immune cells recruitment:** Furthermore, in an *in vivo* system, the presence of Tumor-Infiltrating Lymphocytes (TILs) is not solely dependent on the proliferation or maturation of existing immune cells in the tumor but is also influenced by the infiltration of immune cells from distant sites, such as those from draining lymph nodes or the spleen. The Pathogen-Associated Molecular Patterns (PAMPs) from *E. coli* and Damage-Associated Molecular Patterns (DAMP) from oxaliplatin-treated MC38 both served as are potent immune attractants that are capable of stimulating the proliferation of immune cells in secondary lymphoid organs and facilitating their migration into the tumor microenvironment (TME).
- This additional mechanism underscores the multifaceted impact of *E. coli* in our study, as it not only supports the maintenance and proliferation of immune cells within the TME but also promotes the recruitment of immune cells from peripheral sources, contributing to the observed increase in TILs. Further investigations could delve into the precise pathways and signaling mechanisms involved in this immune cell mobilization and infiltration, thereby enhancing our understanding of these complex interactions.

Figure R3. Counts of CD4+ T-cells, CD8+ T-cells, B-cells, dendritic cells and macrophages in the MC38-spleenocytes co-culture assay. (NT: no treatment; M: MC38-spleenocytes; MO: MC38-spleenocytes with 2 μ M oxaliplatin; ME: MC38-spleenocytes with 2E6 *E. coli*; MOE: MC38-spleenocytes with 2 μ M oxaliplatin and 2E6 *E. coli*)

7. Results: Fig. 4a-d. Only TNF-a and IFN-g is used as activation marker. Additional markers including other activation markers need to be included in the analysis.

Response: We thank the reviewer for the valuable comment. While we acknowledge that a wide arrays of T cell activation markers (i.e. CD25, CD44, CD107a, CD137...etc) could add to the rigor of the tumor infiltrating lymphocyte analysis, we would like to clarify that for the purpose of the present article, TNF-a and IFN-g are sufficient in portraying the increasing number of functional tumor infiltrating lymphocytes

in the TME following oxa+E. coli therapy. In conjunction with Figure 2F that proves anticancer immunological memory and Figure 4A, B that show heightened CD4 and CD8 populations in the tumor of oxa+E. coli-treated mice, the putative functional markers of TNF- α and IFN- γ add to our overarching narrative of enhancing immune cell populations to exert differential selective pressure on cancer cells. We acknowledge that our original manuscript had a liberal use of “activation” when describing TILs, which led the reviewer to seek for additional activation markers. As such, we revised the descriptions to “functional” T cells when describing TNF α + and IFN γ + CD8 T cells in the TME.

8. Results: Fig. 5e. Legend is missing. It is completely unclear what is shown here.

- **Response:** We thank the reviewer for raising this issue, the color codes were added back to the figure.

9. Results in vitro: as pointed out already, these data are extremely preliminary. Fig. 5. These data are meaningless unless the absolute numbers are shown. Numbers of innate immune cells like DC or macrophages increase. Are they proliferating? This is unlikely since they are considered to be end cells. Under some conditions, they might proliferate. This needs to be shown. Why was isolated LPS not tested?

Response: The objective of the *in vitro* assay is to supplement our *in vivo* data by investigating the potential interactions among each treatment arm in a controlled environment and provide a hypothesis of how those interactions contribute to the cancer cell cytotoxicity. Our *in vitro* data showed that E. coli is responsible for the activation of immune cells (both APCs and T-cells; Figure R4B, C and D; included as Figure 5B, C and D in the manuscript body respectively) while oxaliplatin bridged the activated immune cells to cancer cell cytotoxicity (Figure R4E; included as Figure 5E in the manuscript body). We appreciate the reviewer's guidance and have taken the necessary steps to address the concerns raised:

- **Presentation of *In Vitro* Spleenocyte Assays:** We have re-conducted the *in vitro* spleenocyte assays and have now presented the data in Figure R4 in the form of cell number/mL (included as Figure 5 in the manuscript body), as suggested.
- **Dendritic Cells (DC) and Macrophages:** While DCs and macrophages may not typically proliferate in this context, it is indeed possible for monocytes to undergo maturation and transformation into DC or macrophage-like cells, as indicated by the upregulation of maturation markers such as CD11c and F4/80 upon oxaliplatin treatment in the presence of LPS. We appreciate the reference provided to support this observation [8].
- **Cytotoxicity Assay:** We conducted a cytotoxicity assay by co-incubating different concentrations of LPS with oxaliplatin and MC38. However, we observed no significant difference in oxaliplatin-induced cytotoxicity, even at high LPS concentrations (1000 ng/mL; typical *in vitro* dosage = 1-100 ng/mL). (Figure R5A; included as Figure S13D in the manuscript's supplementary data).
- Notably, the addition of whole bacteria (MOI = 10) led to a twofold increase in oxaliplatin potency. This finding suggests that the synergistic effect between *E. coli* and oxaliplatin is likely attributed to factors beyond LPS, such as other Pathogen-Associated Molecular Patterns (PAMPs) from the bacteria, including flagellin, CpG ODN, peptidoglycan, and more. These additional PAMPs may play a crucial role in enhancing the combined therapeutic efficacy of *E. coli* and oxaliplatin. (Figure R5B; included as Figure S13E in the manuscript's supplementary data).

- We believed that these adjustments and explanations will address the concerns raised and provide a clearer understanding of our research findings.

Figure R4. The cancer/immune/bacteria co-culture system reveals the synergistic interaction of OXA and *E. coli* for immune cell activation. The immune cells (murine splenocytes) and the cancer cells (MC38) were co-cultured and subject to various treatments (NT: splenocytes alone. M: splenocytes + MC38. ME: splenocytes + MC38 + *E. coli*. MO: splenocytes + MC38 + oxaliplatin. MOE: splenocytes + MC38 + oxaliplatin + *E. coli*). (A) The populations of the three APCs (B cells, DCs, and macrophages) in the CD45⁺ cells. (B) The MHC-II⁺ populations of the three APCs in the CD45⁺ cells. (C) The CD80⁺/CD86⁺ double positive populations of the three APCs in the CD45⁺ cells. (D) The CD25⁺ population in CD8⁺ T cells. (E) Percentage of the dead MC38-EGFP cells after receiving each treatment in a splenocyte co-culture system. Statistical differences between among groups were computed using one-way ANOVA with Turkey's multiple comparison test and differences were considered statistically significance when p-value is < 0.05 (n = 3 for each group).

Figure R5. Cytotoxicity assay (CCK-8) of oxaliplatin towards MC38 in the presence of LPS or *E. coli*. (A) MC38 treated with various concentration of oxaliplatin in presence of 100 ng/mL or 1000 ng/mL LPS. (B) MC38 treated with various concentrations of oxaliplatin in presence of 5E4 and 5E5 *E. coli*.

10. Results: the authors then investigate the reactivity of spleen cells from naïve mice towards the tumor cells. First of all, it is not clear how they determine cytotoxicity or inhibition in the mixture of tumor and spleen cells. Then they claim that the spleen cells specifically attack the tumor cells when Oxa and *E. coli* are in the co-culture. They try to show this by pictures of aggregates between tumor cells and spleen cells after two days of co-culture without staining the T cells. Do the authors really believe that that many tumor specific T cells are in the spleen of a naïve mouse?

Response: We thank the reviewer for raising this issue. We would like to clarify this issue by showing some of our preliminary data while performing assays optimization and provide some explanations to the questions raised.

- **Whether the CCK8 signals originated from tumor or spleen cells:**

- Prior to CCK8 assay, each well was washed thrice with PBS as most bacteria and spleenocytes are non-adherent (except for those who adhered to the surface of MC38).
- In addition, signal changes in CCK8 assays rely largely on the metabolic activity of the cells studied. Prior to the assays, we conducted an analysis using only spleenocytes to determine its reactivity to the CCK8 reagent. The metabolic activity of the spleenocytes were significantly slower compared to the MC38 and in Figure R6 we showed a standard curve of number of spleenocytes versus A450 signal from the CCK8 assay. We also showed the normalized absorbance values of the co-culture assays with A450 value of 5×10^4 spleenocytes (assuming no spleenocytes were washed away) where the A450 contributed by the spleenocytes were very low and it should not interfere with the results of the CCK8 assay (as Figure R6; included as Figure S13B and S13C in the manuscript's supplementary information).

Figure R6. Assessment of CCK-8 signals for murine spleenocytes. (A) Standard curve demonstrating correlation between spleenocyte counts with the CCK-8 signals (A450). Spleenocytes were counted using hemocytometer under microscope prior to the generation of standard curve. (B) Raw absorbance data for the MC38-spleenocyte co-culture assay. In the co-culture assay, 5×10^4 spleenocytes were added to each

well and the corresponding CCK-8 signals were plotted on the graph with assumption of none of the splenocytes were washed away during the washing steps.

- **Was it growth inhibition or cytotoxic of MC38 cancer cells by the treatment given:**
 - Mix effects were proposed in the combination treatment. At lower concentration (<12.5 uM), oxaliplatin mainly inhibits cancer cell growth without triggering massive cell death.
 - However, in the presence of oxaliplatin, splenocytes and *E. coli*, most of the cancer cells are killed/lysed as a concerted work of all the treatments, evidenced by the live-dead staining of MC38-EGFP cells, where only dead cells (cells with disrupted cell membrane) would be stained (Figure R7; included as Figure 5E in the form of bar graph in the manuscript's body and Fig S15 in the manuscript's supplementary data).

Figure R7. Flow cytometry analysis of the viability of MC38-EGFP cells after 24 h co-culture with murine splenocytes in various conditions. Viability of MC38-EGFP cells were determine using Efluor780 fixable viability dye (Thermofisher Scientific). (M: MC38-splenocytes; MO: MC38-splenocytes with 2 uM oxaliplatin; ME: MC38-splenocytes with 2E6 *E. coli*; MOE: MC38-splenocytes with 2 uM oxaliplatin and 2E6 *E. coli*)

- **Whether if that are so many tumor specific T cells are in the spleen of a naïve mouse:**
 - We agree that the spleen of a naïve mouse should not primarily contain tumor-specific T cells. Our hypothesis is that the observed killing effects of the splenocytes result from the collaborative efforts of both innate and adaptive immune cells.
 - We do not suggest that all the splenocytes adhering to the MC38 cancer cells are CD8+ T cells. Instead, it's likely a combination of activated phagocytes such as macrophages and dendritic cells, along with T cells, actively sampling antigens presented by these phagocytes.
 - When MC38 is treated with oxaliplatin, it can induce immunogenic cell death, leading to processes such as the surface exposure of calreticulin and the release of HMGB1 and ATP, which can attract antigen-presenting cells (APCs) to bind to the cancer cells [9, 10]. Furthermore, our flow cytometry data show that activated APCs upregulate their antigen-presenting capacities (MHC-II) and co-stimulatory receptors (CD80 and CD86). This upregulation could enhance APC-T cell interactions, increasing the likelihood of T cell antigen sampling from APCs. Therefore, it's

possible for cancer-specific T cells to proliferate *in situ* upon receiving antigen presentations (signal 1) and CD80/86 co-stimulation (signal 2) from APCs within the co-culture system.

- We conducted confocal microscopy experiments to identify the types of immune cells that were attached to the MC38 cells. Our observations revealed that dendritic cells, CD8 T-cells, and macrophages were among the immune cell types that bound to the surface of the MC38 cells. (Figure R8; included as Figure S16 in the manuscript's supplementary data).
-

Figure R8. Images of immunofluorescence staining to visualize the interactions between MC38-EGFP cells, spleenocytes, and *E. coli* in the presence of Oxaliplatin (2 µM). The co-culture was first washed, fixed, blocked, and stained with Alexa Fluor® 488 anti-GFP (1:50; Biolegend: 338008) to highlight MC38-EGFP cells. To identify specific immune cell types binding to MC38-EGFP cells, we individually stained the samples with rabbit anti-mouse CD8a (1:50 Abcam: ab217344), rabbit anti-mouse CD11c (1:50 Abcam: ab219799), and rabbit anti-mouse F4/80 (1:50 Abcam: ab300421). These primary antibodies were incubated overnight, followed by staining with secondary antibodies (1:2000; Goat anti-rabbit CF-568 Biotum 20102) and Hoechst 33342 (1:10000; Invitrogen) for an hour before visualization using confocal microscopy (ZEISS Airyscan; LSM880). It's worth noting that we attempted to stain CD4-T cells and B-lymphocytes using anti-mouse CD4 Alexa 647 (1:50; BD Pharmingen 557681) and anti-mouse CD19 Alexa 647 (1:50; BD Pharmingen 557684), but no binding of these cell types to the MC38-EGFP cells was observed in the samples.

11. Discussion: it is not true that bacteria do not support the adaptive immune response against tumors. This has been shown in several publications.

- **Response:** We appreciate the reviewer's feedback and have dialed back our tune in the revised manuscript. We have removed statements indicating that bacteria does not support the adaptive immune response against tumors. In the revised manuscript, we highlight that the combination of bacteria and immunogenic chemotherapy can bolster tumor-reactive adaptive immune responses.

12. Supplement: in general, all the flow cytometry panels are too small. The writing here is most of the time unreadable.

- **Response:** We thank the reviewer for pointing out this issue, we had remade the graphics in landscape format and provided images with better resolution to ensure the labels on the flow cytometry panels are readable upon enlargement of the figures.

13. Minor points: the English of the manuscript is well readable. Only at a few locations the wording is miss-understandable. Results: 7. Line: "than that for E. coli" (not of). Page 8, last line: "monotreatment" not single treatment since the bacteria have been injected several times.

- **Response:** We thank the reviewer for raising these issues, we had made the corresponding amendment as per suggested.

Reference(s) :

1. Chart, H., et al., *An investigation into the pathogenic properties of Escherichia coli strains BLR, BL21, DH5alpha and EQ1*. Journal of applied microbiology, 2000. **89**(6): p. 1048-1058.
2. Chowdhury, S., et al., *Programmable bacteria induce durable tumor regression and systemic antitumor immunity*. Nature Medicine, 2019. **25**(7): p. 1057-1063.
3. Chiang, C.-J. and P.-H. Huang, *Metabolic engineering of probiotic Escherichia coli for cytolytic therapy of tumors*. Scientific Reports, 2021. **11**(1): p. 5853.
4. Buerfent, B.C., et al., *Escherichia coli-induced immune paralysis is not exacerbated during chronic filarial infection*. Immunology, 2015. **145**(1): p. 150-160.
5. Chiang, C.J. and Y.H. Hong, *In situ delivery of biobutyrate by probiotic Escherichia coli for cancer therapy*. Sci Rep, 2021. **11**(1): p. 18172.
6. Miao, H., et al., *Quantifying the early immune response and adaptive immune response kinetics in mice infected with influenza A virus*. J Virol, 2010. **84**(13): p. 6687-98.
7. D'Orazio, S.E.F., *Innate and Adaptive Immune Responses during Listeria monocytogenes Infection*. Microbiol Spectr, 2019. **7**(3).
8. Kim, N.-R. and Y.-J. Kim, *Oxaliplatin regulates myeloid-derived suppressor cell-mediated immunosuppression via downregulation of nuclear factor- κ B signaling*. Cancer Medicine, 2019. **8**(1): p. 276-288.
9. Tesniere, A., et al., *Immunogenic death of colon cancer cells treated with oxaliplatin*. Oncogene, 2010. **29**(4): p. 482-491.
10. Obeid, M., et al., *Calreticulin exposure dictates the immunogenicity of cancer cell death*. Nat Med, 2007. **13**(1): p. 54-61.

28th Nov 2023

Dear Dr. Hu,

Thank you for the submission of your revised manuscript to EMBO Molecular Medicine. We have now heard back from the two referees who agreed to re-evaluate your manuscript. As you will see from the report below, while the referee #1 is supporting publication of the manuscript, the referee #3 acknowledges the improvements of the revised manuscript but also raises some concerns that should be addressed in additional and final round of revision. Specifically, please consider removing the figures and results as indicated in point #3 and point #7.

Acceptance or rejection of the manuscript will depend on the completeness of your responses included in the next, final version of the manuscript. For this reason, and to save you from any frustrations in the end, I would strongly advise against returning an incomplete revision.

In addition, please amend the following:

1) Figures:

- We note that some images/panels are reused. Figure 2D is reused in Appendix Figure S3 and Figure 6D in Appendix Figure S14. Please cite in the respective figure legend every reused image/panel.
- In figures where $n=2$ please show raw values from both measurements and remove error bars. When $n=2$ statistical analysis is not recommended and a justification for the use of the statistical test employed has to be provided. Please check our Author Guidelines:

<https://www.embopress.org/page/journal/17574684/authorguide#statisticalanalysis>

2) In the main manuscript file, please do the following:

- Please address all comments suggested by our data editors listed below:

- o Please note that the error bars are not defined in the legend of figures 1a-c, e-g; 2a-c, f-h; 3a-j; 4a-d, f-h; 5a-e; 6a-b.

- o Please note that $n=2$ in figures 3b, c, e, f, i, j; 4c-d.

- o Please note that the figure legend style does not comply with the journal guidelines i.e. all the figure legends are in a run-on style.

- o Please note that a separate 'Data Information' section is required in the legends of all the figures.

- o Please define the annotated p values ****/**/**/* in the legend of figures 1c, g; 2c, f; 3a-b, d-j; 4a-c, f-h; 5a-e; 6a-b as appropriate.

- Limit keywords to max 5.

- Please add callout for Fig 1H, 2H and 3F.

- In M&M, provide the antibody dilutions that were used for each antibody.

- Please rename "Conflict of Interest" to "Disclosure Statement & Competing Interests". We updated our journal's competing interests policy in January 2022 and request authors to consider both actual and perceived competing interests. Please review the policy <https://www.embopress.org/competing-interests> and update your competing interests if necessary.

- Please correct the reference citation in the text and reference list. In the text of the manuscript, a reference should be cited by author and year of publication. Include a space between a word and the opening parenthesis of the reference that follows. In the reference list, citations should be listed in alphabetical order. Where there are more than 10 authors on a paper, 10 will be listed, followed by "et al.". Please check "Author Guidelines" for more information.

<https://www.embopress.org/page/journal/17574684/authorguide#referencesformat>

3) Appendix: Please add table of content with page numbers and correct nomenclature for Appendix Figure S1 etc., also in the main text.

4) Funding: Please make sure that information about all sources of funding are complete in both our submission system and in the manuscript. Currently there is a discrepancy between entry in our submission system and manuscript: National Science and Technology Council of the Republic of China in the manuscript file and Ministry of Science and Technology, Taiwan (MOST) in our system. Please correct.

5) The Paper Explained: Please provide "The Paper Explained" and add it to the main manuscript text. Please check "Author Guidelines" for more information. <https://www.embopress.org/page/journal/17574684/authorguide#researcharticleguide>

6) Synopsis: Every published paper now includes a 'Synopsis' to further enhance discoverability. Synopses are displayed on the journal webpage and are freely accessible to all readers. They include separate synopsis image and synopsis text.

- Synopsis image: Please provide a striking image or visual abstract as a high-resolution jpeg file 550 px-wide x (250-400)-px high to illustrate your article.

- Synopsis text: Please provide a short standfirst (maximum of 300 characters, including space) as well as 2-5 one sentence bullet points that summarise the paper as a .doc file. Please write the bullet points to summarise the key NEW findings. They should be designed to be complementary to the abstract - i.e. not repeat the same text. We encourage inclusion of key acronyms and quantitative information (maximum of 30 words / bullet point). Please use the passive voice.

7) For more information: This space should be used to list relevant web links for further consultation by our readers. Could you identify some relevant ones and provide such information as well? Some examples are patient associations, relevant databases,

OMIM/proteins/genes links, author's websites, etc...

8) As part of the EMBO Publications transparent editorial process initiative (see our Editorial at <http://embomolmed.embopress.org/content/2/9/329>), EMBO Molecular Medicine will publish online a Review Process File (RPF) to accompany accepted manuscripts. This file will be published in conjunction with your paper and will include the anonymous referee reports, your point-by-point response and all pertinent correspondence relating to the manuscript. Let us know whether you agree with the publication of the RPF and as here, if you want to remove or not any figures from it prior to publication. Please note that the Authors checklist will be published at the end of the RPF.

9) Please provide a point-by-point letter INCLUDING my comments as well as the reviewer's reports and your detailed responses (as Word file).

I look forward to reading a new revised version of your manuscript as soon as possible.

Yours sincerely,

Zeljko Durdevic

*** Instructions to submit your revised manuscript ***

- 1) a .docx formatted version of the manuscript text (including Figure legends and tables)
- 2) Separate figure files*
- 3) supplemental information as Expanded View and/or Appendix. Please carefully check the authors guidelines for formatting Expanded view and Appendix figures and tables at <https://www.embopress.org/page/journal/17574684/authorguide#expandedview>
- 4) a letter INCLUDING the reviewer's reports and your detailed responses to their comments (as Word file).
- 5) The paper explained: EMBO Molecular Medicine articles are accompanied by a summary of the articles to emphasize the major findings in the paper and their medical implications for the non-specialist reader. Please provide a draft summary of your article highlighting
 - the medical issue you are addressing,
 - the results obtained and
 - their clinical impact.This may be edited to ensure that readers understand the significance and context of the research. Please refer to any of our published articles for an example.
- 6) For more information: There is space at the end of each article to list relevant web links for further consultation by our readers. Could you identify some relevant ones and provide such information as well? Some examples are patient associations, relevant

databases, OMIM/proteins/genes links, author's websites, etc...

7) Author contributions: the contribution of every author must be detailed in a separate section.

8) EMBO Molecular Medicine now requires a complete author checklist (<https://www.embopress.org/page/journal/17574684/authorguide>) to be submitted with all revised manuscripts. Please use the checklist as guideline for the sort of information we need WITHIN the manuscript. The checklist should only be filled with page numbers where the information can be found. This is particularly important for animal reporting, antibody dilutions (missing) and exact values and n that should be indicated instead of a range.

9) Every published paper now includes a 'Synopsis' to further enhance discoverability. Synopses are displayed on the journal webpage and are freely accessible to all readers. They include a short stand first (maximum of 300 characters, including space) as well as 2-5 one sentence bullet points that summarise the paper. Please write the bullet points to summarise the key NEW findings. They should be designed to be complementary to the abstract - i.e. not repeat the same text. We encourage inclusion of key acronyms and quantitative information (maximum of 30 words / bullet point). Please use the passive voice. Please attach these in a separate file or send them by email, we will incorporate them accordingly.

You are also welcome to suggest a striking image or visual abstract to illustrate your article. If you do please provide a jpeg file 550 px-wide x 300-800px high.

10) A Conflict of Interest statement should be provided in the main text

11) Please note that we now mandate that all corresponding authors list an ORCID digital identifier. This takes <90 seconds to complete. We encourage all authors to supply an ORCID identifier, which will be linked to their name for unambiguous name identification.

Currently, our records indicate that the ORCID for your account is 0000-0002-0988-7029.

Link Not Available

Photos 400-800 DPI

*Additional important information regarding figures and illustrations can be found at <https://bit.ly/EMBOPressFigurePreparationGuideline>. See also figure legend preparation guidelines: <https://www.embopress.org/page/journal/17574684/authorguide#figureformat>

***** Reviewer's comments *****

Referee #1 (Remarks for Author):

The authors have improved the manuscript massively and it reads more fluent and clearer now. From my point of view, the manuscript is now suitable for publication in EBMP Mol Med.

Referee #3 (Comments on Novelty/Model System for Author):

I find this points satisfactory in the manuscript

Referee #3 (Remarks for Author):

In this revised version, Lim et al. describe the synergy of Oxaliplatin and E. coli for tumor therapy. The revision has significantly improved the manuscript. Nevertheless some points are still problematic. In detail (note minor and major points are mixed):

1. English: in general English is good but at some locations it needs to be corrected, like: 1 line Introduction:...comprising of cancer cells..., or Introduction:...mice models...etc.
2. Results: it needs to be mentioned at the beginning of the results that the application of E. coli is carried out intratumorally . This is important because in most publication bacteria are applied i.v.
3. Results: the starvation together with FU and E. coli does not any significant effect on tumor volume. Nevertheless the authors show a Kaplan-Mayer-Blot where the survival which is based on tumor size and by which the authors try to proof a high efficacy of the combinatorial treatment. However, they stop the experiment at day 16. Thus it is not clear which size the tumors would have had a day later. Since the experiment is apparently carried out only once, it is also not clear how reproducible the finding is. Anyway, I do not understand why the authors display this experiment since in the subsequent experiments they do not apply this condition. It should be deleted as figure and in the text.
4. Results: the authors claim that the immune response in tumors experimentally colonized by bacteria is mainly directed against the bacteria. I am not aware of a study showing this.
5. Results: The authors claim that the absence of enhancement of T cell infiltration in Tumors treated with FU+E. coli is expected, but quote only papers on FU.
6. Results: in the rechallenge experiments, the authors do not mention after which time the rechallenge was done. No control unrelated tumor was applied. Although, are most likely corrected they cannot exclude the mice were in a general inflammatory state that does not allow to implant subcutaneously a tumor.
7. Results: the increase of cytotoxicity when spleen cells are added is not very impressive and I am not convinced that it is meaningful. How often was the experiment done? What are the ideas of the authors about the mechanism that provides the additional cytotoxicity.

Response to Editor:

1) Figures:

- We note that some images/panels are reused. Figure 2D is reused in Appendix Figure S3 and Figure 6D in Appendix Figure S14. Please cite in the respective figure legend every reused image/panel.
- In figures where $n=2$ please show raw values from both measurements and remove error bars. When $n=2$ statistical analysis is not recommended and a justification for the use of the statistical test employed has to be provided. Please check our Author Guidelines: <https://www.embopress.org/page/journal/17574684/authorguide#statisticalanalysis>

Response:

- We thank the editor for the information. We had included the respective description in the legends of Appendix Figure S3 to indicate that part of the figures was presented in the main text.
 - *“Part of the figures were presented in Figure 2D.”*
 - For Figure 6D, the figure was removed from the manuscript according to the reviewer’s advice.
 - For figures with $n = 2$, we had removed all the error bars and present the data as individual plot for easy visualization.
- 2) In the main manuscript file, please do the following:
- Please address all comments suggested by our data editors listed below:
 - o Please note that the error bars are not defined in the legend of figures 1a-c, e-g; 2a-c, f-h; 3a-j; 4a-d, f-h; 5a-e; 6a-b.

Response:

- The following sentence is added into the figure legend for definition of the data plot and error bar:
 - o For Figure 1 and 2,: *“All curves are presented as mean with SEM as error bars and all column plots are presented as individual value with SEM error bars.”*
 - o For Figure 3 and 4: *“All data were presented as individual data plots and error bars were presented as SEM of the data set if applied.”*
 - o For Figure 5: *“All data were presented as individual data plots with SEM error bars.”*
- o Please note that $n=2$ in figures 3b, c, e, f, i, j; 4c-d.
- For figures with $n = 2$, we had removed all the error bars and present the data as individual plot for easy visualization.
- o Please note that the figure legend style does not comply with the journal guidelines i.e. all the figure legends are in a run-on style.
- We had reformatted all the figure legends in the main text to comply with the journal requirement. An example of the reformatted legend is as follow:

“Figure 1. 5-FU or starvation provided differential selection pressure against cancer cells over bacteria.

A. *In vitro* cytotoxicity of 5-FU against the CRC cancer cell line (MC38; n = 3).

B. *In vitro* cytotoxicity of 5-FU against the *E. coli* (DH5 α ; n = 3).

C. *In vivo* anti-tumor activity of PBS, 5-FU, *E. coli*, and 5-FU+*E. coli* in the MC38 syngeneic murine model, 5-FU and *E. coli* were delivered intraperitoneally and intratumorally respectively (n = 5 mice for each treatment group).

D. Kaplan-Meier analysis of mouse survivals in (C).

*Data information: Mice were considered dead when the tumor volume exceeded 1,500 mm³. All curves are presented as mean with SEM as error bars and all column plots are presented as individual value with SEM error bars. Statistical differences among tumor-growth for more than 2 groups were computed using two-way ANOVA with Turkey’s multiple comparison test. The differences were considered statistically significant when p-value < 0.05 in all statistical analysis. Significance symbols are defined as * p < 0.05; ** p = 0.01 – 0.05; *** p = 0.0001 – 0.001; and **** p < 0.0001.”*

o Please note that a separate 'Data Information' section is required in the legends of all the figures.

- A separate data information section is included in the legends of Figure 1 – 6. No additional data information to disclose in Figure 7 (illustration of idea).

o Please define the annotated p values ****/**/*/* in the legend of figures 1c, g; 2c, f; 3a-b, d-j; 4a-c, f-h; 5a-e; 6a-b as appropriate.

- Symbols of annotated p-values are defined in the figure legends as requested: “Significance symbols are defined as * p < 0.05; ** p = 0.01 – 0.05; *** p = 0.0001 – 0.001; and **** p < 0.0001. ”

- Limit keywords to max 5.

- The list of keywords is reduced to 5: “Keywords: bacteria cancer therapy, immunotherapy, differential stress resistance, oxaliplatin, TME remodeling”.

- Please add callout for Fig 1H, 2H and 3F.

- Statistical callout had been included in Figure 2H. However, Figure 1H have been removed from the manuscript body according to reviewer suggestion. For Figure 3F, callout was initially included in the figure, but it was taken out in accordance to the manuscript policy that suggests statistical analysis should not performed when n=2 (as in point no. 1).

- In M&M, provide the antibody dilutions that were used for each antibody.

- We had included the dilution factors for all antibodies listed in the material and method section. An example is as follow:

- “eFluor™ 780 eBioscience™ Fixable Viability Dye (Thermofisher Scientific; Cat# 65-0865; 1: 2000)”

- Please rename "Conflict of Interest" to "Disclosure Statement & Competing Interests". We updated our journal's competing interests policy in January 2022 and request authors to consider both actual and perceived competing interests. Please review the policy <https://www.embopress.org/competing-interests> and update your competing interests if necessary.

- We had renamed the section to Disclosure Statement & Competing Interests as requested.

- Please correct the reference citation in the text and reference list. In the text of the manuscript, a reference should be cited by author and year of publication. Include a space between a word and the opening parenthesis of the reference that follows. In the reference list, citations should be listed in alphabetical order. Where there are more than 10 authors on a paper, 10 will be listed, followed by "et al.". Please check "Author Guidelines" for more information. <https://www.embopress.org/page/journal/17574684/authorguide#referencesformat>

- All citations have been updated to comply with the journal requirement (author, date) and the reference section was also modified according to the guidelines given.

3) Appendix: Please add table of content with page numbers and correct nomenclature for Appendix Figure S1 etc., also in the main text.

- A table of content was added as requested and all the figures in appendix were renamed into Appendix Figure 1 ... 16 accordingly, in the main text and appendix section.

4) Funding: Please make sure that information about all sources of funding are complete in both our submission system and in the manuscript. Currently there is a discrepancy between entry in our submission system and manuscript: **National Science and Technology Council of the Republic of China** in the manuscript file and Ministry of Science and Technology, Taiwan (MOST) in our system. Please correct.

- We apologize for the misunderstanding. There is a recent change in the naming of the funding agency to **National Science and Technology Council of the Republic of China**. We had made the relevant correction in the manuscript submission system.

5) The Paper Explained: Please provide "The Paper Explained" and add it to the main manuscript text. Please check "Author Guidelines" for more information. <https://www.embopress.org/page/journal/17574684/authorguide#researcharticleguide>

- A “The Paper Explained” section is added into the end of the main text, addressing the problem, result and impact of the study.

6) Synopsis: Every published paper now includes a 'Synopsis' to further enhance discoverability. Synopses are displayed on the journal webpage and are freely accessible to all readers. They include separate synopsis image and synopsis text.

- Synopsis image: Please provide a striking image or visual abstract as a high-resolution jpeg file 550 px-wide x (250-400)-px high to illustrate your article.

- Synopsis text: Please provide a short standfirst (maximum of 300 characters, including space) as well as 2-5 one sentence bullet points that summarise the paper as a .doc file. Please write the bullet points to summarise the key NEW findings. They should be designed to be complementary to the abstract - i.e. not repeat the same text. We encourage inclusion of key acronyms and quantitative information (maximum of 30 words / bullet point). Please use the passive voice.

- A synopsis figure and word file with synopsis text is included along with the submitted manuscript.

7) For more information: This space should be used to list relevant web links for further consultation by our readers. Could you identify some relevant ones and provide such information as well? Some examples are patient associations, relevant databases, OMIM/proteins/genes links, author's websites, etc...

- Author's Website
- <https://www.ibms.sinica.edu.tw/yun-mou/>

8) As part of the EMBO Publications transparent editorial process initiative (see our Editorial at <http://embomolmed.embopress.org/content/2/9/329>), EMBO Molecular Medicine will publish online a Review Process File (RPF) to accompany accepted manuscripts. This file will be published in conjunction with your paper and will include the anonymous referee reports, your point-by-point response and all pertinent correspondence relating to the manuscript. Let us know whether you agree with the publication of the RPF and as here, if you want to remove or not any figures from it prior to publication. Please note that the Authors checklist will be published at the end of the RPF.

Response:

- We agree to the publication of the RPF files.

9) Please provide a point-by-point letter INCLUDING my comments as well as the reviewer's reports and your detailed responses (as Word file).

Response to Reviewer 1:

Referee #1 (Remarks for Author):

The authors have improved the manuscript massively and it reads more fluent and clearer now. From my point of view, the manuscript is now suitable for publication in EBMP Mol Med.

Response:

- We thank the reviewer for the constructive comments that help us to improve the paper.

Response to Reviewer 3:

Referee #3 (Comments on Novelty/Model System for Author):

I find this points satisfactorial in the manuscript

Referee #3 (Remarks for Author):

In this revised version, Lim et al. describe the synergy of Oxaliplatin and E. coli for tumor therapy. The revision has significantly improved the manuscript. Nevertheless some points are still problematic. In detail (note minor and major points are mixed):

1. English: in general English is good but at some locations it needs to be corrected, like: 1 line Introduction:...comprising of cancer cells..., or Introduction:...mice models...etc.

Response:

- We thank the reviewer for the constructive comments. We had made the relevant corrections as follow:
- **From:** comprising of cancer cells... :
 - **To:** Solid tumors are a complex ecosystem composed of various cell types, including cancer cells ...
- **From:** Mice model:
 - **To:** mouse model

We have also proofread the manuscript to remove other grammatical glitches.

2. Results: it needs to be mentioned at the beginning of the results that the application of E. coli is carried out intratumorally . This is important because in most publication bacteria are applied i.v.

Response:

- We thank the reviewer for the suggestions and we acknowledge the need to stress on the administration mode of the bacteria in our study. We had amended the subtopic at the beginning of the result into :
- *“Differential selection pressure provided by intratumoral bacterial therapy in combination with chemotherapy and fasting inhibited tumor growth.”*

- We also added a descriptive text in the main text to highlight the administration route of the bacteria treatments.
- “*All bacteria treatments were administered intratumorally unless otherwise specified.*”

3. Results: the starvation together with FU and *E. coli* does not any significant effect on tumor volume. Nevertheless the authors show a Kaplan-Mayer-Blot where the survival which is based on tumor size and by which the authors try to proof a high efficacy of the combinatorial treatment. However, they stop the experiment at day 16. Thus it is not clear which size the tumors would have had a day later. Since the experiment is apparently carried out only once, it is also not clear how reproducible the finding is. Anyway, I do not understand why the authors display this experiment since in the subsequent experiments they do not apply this condition. It should be deleted as figure and in the text.

- **Response:**

- We appreciate the reviewer's comments. After a thorough review of our manuscript, we have decided to relocate the starvation results to the supplementary data. Additionally, we have revised the relevant paragraph to emphasize the limitations of the *in vivo* starvation approach, providing a more accurate reflection of the results. We also offer explanations to the audience regarding the decision not to continue with starvation in the subsequent parts of the study.

4. Results: the authors claim that the immune response in tumors experimentally colonized by bacteria is mainly directed against the bacteria. I am not aware of a study showing this.

Response:

- We thank the reviewer for raising this question, we acknowledged the ambiguity in the sentence, and we made the following amendment.
 - **From:** “*the resulting selective pressure is unspecific and primarily targets the bacteria rather than the tumor cells*”
 - **To:** “*the resulting selective pressure is non-tumor specific*”
- The statement above refers to publications highlighting the efficacy of bacteria therapy, indicating that bacteria administration into tumors mainly activates strong infiltrations of innate immune cells such as neutrophils (Westphal et al., 2008) and macrophages (Weibel et al., 2008). These infiltrations lead to both the necrosis of the tumor tissue, bacteria clearance at the proliferating tumor rim as well as containment of the bacteria in the necrotic zone of the tumor. The strong infiltration of innate immune cells could result in temporally suppression of the tumor but also limit the long term therapeutic efficacy of the bacteria monotherapy.
- These observations align with our *in vivo* data, where synergism was observed only between the anti-tumor efficacy of *E. coli* and oxaliplatin. Tumor-infiltrating lymphocyte (TIL) analysis also revealed the highest proportion of neutrophils in the group receiving *E. coli* monotherapy, but not in the combined regimen where very limited numbers of neutrophils were determined and CD8+ T cells dominated the tumor microenvironment (TME).

5. Results: The authors claim that the absence of enhancement of T cell infiltration in Tumors treated with FU+E. coli is expected, but quote only papers on FU.

Response:

- We thank the reviewer for raising this question, we acknowledged the ambiguity in the sentence, and we made the following amendment to ensure clarity of the statements being claimed.

From: *“It is worth noting that 5-FU is a low-immunogenic agent as it could not efficiently induce ICDs in cancer cells (Vincent et al., 2010; Yamamura et al., 2015). Indeed, we found no enhancement of CD4⁺ or CD8⁺ tumor-infiltrating lymphocytes (TILs) in the group of 5-FU+E. coli co-treatment when compared to the other three treatment groups (Fig. S3).”*

To: *“Examination of the TME of the 5-FU+E. coli group showed no enhancement of CD4⁺ or CD8⁺ tumor-infiltrating lymphocytes (TILs) (Appendix Figure S3), indicating that the treatment combination was ineffective in elevating anticancer adaptive immunity (Vincent et al., 2010; Yamamura et al., 2015).”*

6. Results: in the rechallenge experiments, the authors do not mention after which time the rechallenge was done. No control unrelated tumor was applied. Although, are most likely corrected they cannot exclude the mice were in a general inflammatory state that does not allow to implant subcutaneously a tumor.

Response:

- We appreciate the reviewer for bringing up this concern and apologize for overlooking the details of the rechallenge in the manuscript. We have now included the relevant information on the rechallenge experiment in the methodology section as follows:
 - *“The rechallenge was performed at least 3 weeks after the last administration of E. coli and two weeks after no tumor growth was observed in mice.”*
- While we acknowledge the possibility raised by the reviewer, we consider it unlikely for the general inflammation stage to persist for such a prolonged period that hinders tumor growth. Moreover, studies have indicated that chronic inflammation promotes all stages of tumorigenesis.

7. Results: the increase of cytotoxicity when spleen cells are added is not very impressive and I am not convinced that it is meaningful. How often was the experiment done? What are the ideas of the authors about the mechanism that provides the additional cytotoxicity.

- We appreciate the reviewer’s feedback and after careful consideration, we decided to move the relevant figures into the supplementary data section.
- The experiment was done twice (biological replicate) with at least three replicates for each data point. We hypothesized that the additional toxicity from splenocytes could be attributed to the following reasons:
 - Oxaliplatin induce immunogenic cell death in MC38, leading to processes such as the surface exposure of calreticulin and the release of HMGB1 and ATP, which can attract antigen-presenting cells (APCs) to bind to the cancer cells (Alcindor & Beauger, 2011; Stojanovska et al., 2019; Tesniere et al., 2010).
 - APCs such as macrophages could be activated/polarized into M1-state that could exert their tumoricidal activity via secretion of reactive nitrogen/oxygen species as well as pro-inflammatory cytokines (Boutillier & ElSawa, 2021; Guo et al., 2017; Li et al., 2021).
 - Direct cytotoxicity from APC-primed T-cells to MC38 cells.
 - We also conducted confocal microscopy experiments to identify the types of immune cells that were attached to the MC38 cells. Our observations revealed that dendritic cells,

CD8 T-cells, and macrophages were among the immune cell types that bound to the surface of the MC38 cells. (Figure S16 in the manuscript's supplementary data).

Reference(s):

- Alcindor, T., & Beauger, N. (2011). Oxaliplatin: a review in the era of molecularly targeted therapy. *Current oncology (Toronto, Ont.)*, 18(1), 18-25. <https://doi.org/10.3747/co.v18i1.708>
- Boutillier, A. J., & Elswa, S. F. (2021). Macrophage Polarization States in the Tumor Microenvironment. *Int J Mol Sci*, 22(13). <https://doi.org/10.3390/ijms22136995>
- Guo, B., Li, L., Guo, J., Liu, A., Wu, J., Wang, H., Shi, J., Pang, D., & Cao, Q. (2017). M2 tumor-associated macrophages produce interleukin-17 to suppress oxaliplatin-induced apoptosis in hepatocellular carcinoma. *Oncotarget*, 8(27), 44465-44476. <https://doi.org/10.18632/oncotarget.17973>
- Li, F., Zheng, X., Wang, X., Xu, J., & Zhang, Q. (2021). Macrophage polarization synergizes with oxaliplatin in lung cancer immunotherapy via enhanced tumor cell phagocytosis. *Transl Oncol*, 14(11), 101202. <https://doi.org/10.1016/j.tranon.2021.101202>
- Stojanovska, V., Prakash, M., McQuade, R., Fraser, S., Apostolopoulos, V., Sakkal, S., & Nurgali, K. (2019). Oxaliplatin Treatment Alters Systemic Immune Responses. *Biomed Res Int*, 2019, 4650695. <https://doi.org/10.1155/2019/4650695>
- Tesniere, A., Schlemmer, F., Boige, V., Kepp, O., Martins, I., Ghiringhelli, F., Aymeric, L., Michaud, M., Apetoh, L., Barault, L., et al. (2010). Immunogenic death of colon cancer cells treated with oxaliplatin. *Oncogene*, 29(4), 482-491. <https://doi.org/10.1038/onc.2009.356>
- Vincent, J., Mignot, G., Chalmin, F., Ladoire, S., Bruchard, M., Chevriaux, A., Martin, F., Apetoh, L., Rebe, C., & Ghiringhelli, F. (2010). 5-Fluorouracil selectively kills tumor-associated myeloid-derived suppressor cells resulting in enhanced T cell-dependent antitumor immunity. *Cancer Res*, 70(8), 3052-3061. <https://doi.org/10.1158/0008-5472.CAN-09-3690>
- Weibel, S., Stritzker, J., Eck, M., Goebel, W., & Szalay, A. A. (2008). Colonization of experimental murine breast tumours by Escherichia coli K-12 significantly alters the tumour microenvironment [<https://doi.org/10.1111/j.1462-5822.2008.01122.x>]. *Cellular Microbiology*, 10(6), 1235-1248. <https://doi.org/https://doi.org/10.1111/j.1462-5822.2008.01122.x>
- Westphal, K., Leschner, S., Jablonska, J., Loessner, H., & Weiss, S. (2008). Containment of tumor-colonizing bacteria by host neutrophils. *Cancer Res*, 68(8), 2952-2960. <https://doi.org/10.1158/0008-5472.Can-07-2984>
- Yamamura, Y., Tsuchikawa, T., Miyauchi, K., Takeuchi, S., Wada, M., Kuwatani, T., Kyogoku, N., Kuroda, A., Maki, T., Shichinohe, T., et al. (2015). The key role of calreticulin in immunomodulation induced by chemotherapeutic agents. *Int J Clin Oncol*, 20(2), 386-394. <https://doi.org/10.1007/s10147-014-0719-x>

21st Dec 2023

Dear Dr. Hu,

We are pleased to inform you that your manuscript is accepted for publication and is now being sent to our publisher to be included in the next available issue of EMBO Molecular Medicine.
